# On the Role of Batch Size in Stochastic Conditional Gradient Methods

**Rustem Islamov** [1][*]   **Roman Machacek** [2]   **Aurelien Lucchi** [1]   **Antonio Silveti-Falls** [3]   **Eduard Gorbunov** [4][†]
**Volkan Cevher** [5][†]

## Abstract

We study the role of batch size in stochastic conditional gradient methods under a $\mu$-Kurdyka–Łojasiewicz ($\mu$-KL) condition. Focusing on momentum-based stochastic conditional gradient algorithms (e.g., Scion), we derive a new analysis that explicitly captures the interaction between stepsize, batch size, and stochastic noise. Our study reveals a regime-dependent behavior: increasing the batch size initially improves optimization accuracy but, beyond a critical threshold, the benefits saturate and can eventually degrade performance under a fixed token budget. Notably, the theory predicts the magnitude of the optimal stepsize and aligns well with empirical practices observed in large-scale training. Leveraging these insights, we derive principled guidelines for selecting the batch size and stepsize, and propose an adaptive strategy that increases batch size and sequence length during training while preserving convergence guarantees. Experiments on NanoGPT are consistent with the theoretical predictions and illustrate the emergence of the predicted scaling regimes. Overall, our results provide a theoretical framework for understanding batch size scaling in stochastic conditional gradient methods and offer guidance for designing efficient training schedules in large-scale optimization.

## 1. Introduction

Large-scale language model training is constrained by a token budget $T$ rather than by a fixed number of optimization steps. In this regime, we face a familiar batch size tradeoff: increasing the batch size $B$ improves hardware utilization, yet beyond a certain scale it can degrade optimization efficiency and hurt generalization (Goyal et al., 2017; Keskar et al., 2017; Smith et al., 2018b; Shallue et al., 2019).

A token budget-aware viewpoint makes this tradeoff explicit. With batch size $B$ and sequence length $S$, the number of parameter updates is $K := \frac{T}{BS}$, and hence $(B, S)$ and the stepsize jointly determine how effectively the token budget is converted into optimization progress. This coupling raises a central question in model training: *how should $(B, S)$ and the stepsize be chosen, and adapted, to optimize performance under a fixed token budget $T$?*

Recent empirical studies have further refined this picture. In particular, critical batch sizes – the point at which scaling $B$ stops being beneficial – appear to scale primarily with the effective data size and only weakly with model size under a fixed token budget (Zhang et al., 2025; Bergsma et al., 2025). Additionally, the critical batch threshold is often stage-dependent, motivating warmup and stage-wise training schedules (Merrill et al., 2025). Taken together, these findings suggest that the batch size should be treated as a dynamic optimization variable rather than a fixed hyperparameter. However, these insights remain largely empirical: they do not provide explicit optimization error laws as functions of $(B, S, T)$, nor do they characterize when increasing batch size becomes provably detrimental under a fixed token budget.

In parallel, hyperparameter transfer frameworks such as $\mu$P have shown that, with appropriate parameterization and initialization, gradient magnitudes can be kept $\Theta(1)$ across model scales, enabling stable training without retuning learning rates (Yang & Hu, 2021; Yang et al., 2021; 2022). However, these results are inherently local: they ensure that individual updates neither explode nor vanish, but do not address how batch size, sequence length, and stepsize should scale *globally* with the token budget.

Our work bridges this gap by showing that hyperparameters that are locally optimal for a given $(B, S, T)$ can become provably suboptimal as the token budget increases, even under $\mu$P-style initialization. To obtain such global scaling laws, we derive an analysis for stochastic condi-

---

[*]Most of this work was done when Rustem Islamov was a visiting student in the group of Prof. Eduard Gorbunov at MBZUAI, UAE. [†]The last two authors share senior authorship. [1]University of Basel, Switzerland [2]University of Bern, Switzerland [3]CentraleSupélec, France [4]MBZUAI, UAE [5]EPFL, Switzerland. Correspondence to: Rustem Islamov <rustem.islamov@unibas.ch>.

*Proceedings of the 43rd International Conference on Machine Learning*, Seoul, South Korea. PMLR 306, 2026. Copyright 2026 by the author(s).

tional gradient (SCG) methods (Pethick et al., 2025a), a projection-free framework that underlies several modern norm-constrained training algorithms. This class of algorithms is closely aligned with modern optimizers such as Muon (Jordan et al., 2024b).

Our analysis is carried out for stochastic optimization (1) under smoothness (A1) in a general norm, norm equivalence (A2), and a $\mu$-Kurdyka–Łojasiewicz ($\mu$-KL) error bound (A3) (Karimi et al., 2016; Bolte et al., 2007). The $\mu$-KL condition is particularly well matched to SCG geometry, as it relates first-order stationarity to suboptimality measured in the dual norm induced by the linear minimization oracle (LMO).

Specializing our convergence bounds to the fixed-token setting $T = KBS$ yields an explicit, non-monotone dependence of the achievable optimization error on the effective batch–sequence scale $BS$. Three regimes emerge: *(i)* a noise-dominated regime where increasing $BS$ improves performance, *(ii)* an intermediate regime where the best achievable error is essentially independent of $BS$, and *(iii)* a large-batch regime where performance deteriorates as $BS$ grows under a fixed token budget.

Balancing the dominant terms yields a *critical* effective batch–sequence–token (BST) scale rule $BS \asymp T^{2/3}$ up to problem-dependent factors that we derive in this work, revealing how curvature, noise, geometry, and error-bound strength shift the optimal operating point. Importantly, our analysis shows that large batch sizes do not inherently degrade performance: when batch size, sequence length, and learning rate are chosen according to our BST scaling rule, large-batch training remains effective and token-efficient. In contrast to $\mu$P, our perspective disentangles local stability, as controlled by parameterization and initialization, from global efficiency, as governed by token-budget–aware optimization.

Our contributions are as follows:

- **Convergence guarantees for momentum SCG under $\mu$-KL.** We establish convergence guarantees for Algorithm 1 under the $\mu$-KL condition (A3) in a general normed geometry, explicitly tracking the effects of momentum, smoothness, and stochastic gradient noise. Our bounds hold *in expectation* under bounded-variance and $L$-smoothness assumptions.

- **A token-budget view of batch, sequence length, and stepsize scaling.** By translating iteration complexity into token complexity via $T = KBS$, we obtain explicit $(B, S, T)$-dependent error laws and identify the *critical* effective batch-sequence scale $BS$ that separates beneficial from harmful scaling.

- **Actionable adaptive scheduling rules.** We turn the theory into concrete recipes for choosing and up-

*dating* $(\beta, B, S)$ during training under a fixed token budget, yielding the scaling relations in Table 1 and a two-stage (and more generally multi-stage) protocol validated empirically on NanoGPT through batch/sequence-length restart experiments and hyperparameter-transfer ablations in Figures 3-5.

Our results complement classical large-batch heuristics such as linear learning rate scaling with warmup (Goyal et al., 2017) and adaptive batch size schedules (Smith et al., 2018b), while offering a *projection-free* viewpoint rooted in conditional gradient geometry. They are also consistent with empirical observations that there exists a largest useful batch size depending on training stage and problem statistics (McCandlish et al., 2018; Shallue et al., 2019), and provide an explicit optimization-side mechanism for the "too-large batch hurts" regime under a fixed token budget.

## 2. Related Works

**Assumptions in SCG methods.** Convergence analyses for stochastic conditional gradient (SCG) (aka Frank–Wolfe) methods and, more broadly, *LMO-based* methods, have been conducted under various assumptions.

**Smoothness.** Most analyses assume standard $L$-smoothness. However, recent works consider relaxed notions, such as $(L_0, L_1)$-smoothness (Zhang et al., 2020) and other extensions beyond global smoothness (Pethick et al., 2025b; Riabinin et al., 2026). Extending our analysis to these generalized smoothness settings is an interesting direction for future work, but it lies beyond the scope of the present paper.

**Structured nonconvexity.** Most prior work considers either general nonconvex or strongly convex objectives, and therefore does not directly target the learning-rate and batch-size scaling effects observed in large-scale training. This limitation motivates our study under structured nonconvexity.

Several recent works study structured nonconvexity for LMO-based or related methods. Yang et al. (2025) provides analysis under a generalized Polyak–Łojasiewicz condition, which recovers our Assumption 3 as a special case. Their method, however, does not use momentum and assumes almost surely affine bounded noise, in contrast to the bounded variance setting considered here.

Kovalev (2025) studies stochastic conditional gradient methods under star-convexity, a condition closely related to the $\mu$-KL condition. However, our work provides empirical evidence supporting the $\mu$-KL condition in large-scale language model training and utilizes this structure to derive a principled BST scaling rule under a fixed token budget. Finally, Riabinin et al. (2026) study an LMO-based method with adaptive layer-wise learning rates under the classical Polyak-Łojasiewicz (PL) condition (Polyak,

1963; Łojasiewicz, 1964), restricted to the deterministic setting without momentum, leaving the stochastic momentum setting open.

**Works on Hyperparameter Transfer.** Transferring hyperparameters (HPs) tuned on small proxy models to large-scale training has become increasingly important as model sizes grow. This line of work was initiated by the $\mu$P framework (Yang & Hu, 2021; Yang et al., 2021; 2022), which enables zero-shot transfer of learning rates across model *width*, and was later extended to other architectural axes, such as depth (Yang et al., 2023; Dey et al., 2025).

Technically, $\mu$P-style analyses focus on parameterizations that ensure gradient magnitudes and parameter updates remain $\mathcal{O}(1)$ around initialization. These analyses assume a fixed number of tokens processed per step and do not characterize optimization behavior when the number of optimization steps is significantly larger than the model width. To reason about the latter regime, we analyze SCG methods under a $\mu$-KL condition and derive convergence guarantees that explicitly depend on the batch size $B$, sequence length $S$, and total token budget $T$. This trajectory-level analysis allows us to characterize how optimization error accumulates as a function of $(B, S, T)$ and to derive principled scaling rules for jointly adapting batch size, sequence length, and stepsize, complementing prior HP transfer works that focus on local, per-step stability that governs the initial training behavior.

Closely related in spirit, (Wang & Aitchison, 2025) study hyperparameter transfer specifically for AdamW by identifying an EMA timescale induced by decoupled weight decay and deriving scaling rules for the weight-decay coefficient across token budget. Our work addresses a complementary question for SCG methods: how the effective batch–sequence scale and Frank–Wolfe stepsize should co-vary with the token budget.

**Batch Size Scheduling.** Adapting batch size during training is a long-standing and practical idea, motivated by both computational and optimization considerations. Increasing batch size can replace learning-rate decay, reduce the number of parameter updates, and improve parallelism (Smith et al., 2018a). However, compared to small-batch training, large batches may lead to a generalization gap and convergence to sharper minima (Keskar et al., 2017).

A complementary empirical view suggests a *critical batch size* (CBS), beyond which increasing $B$ yields diminishing token efficiency; McCandlish et al. (2018) relate CBS to the gradient noise scale and argue that it evolves during training. In the LLM setting, scaling-law work (Kaplan et al., 2020; Hoffmann et al., 2022) primarily addresses how to allocate a fixed compute budget across model size and training tokens, rather than prescribing within-run batch-size schedules. More recently, Bi et al. (2024) report empirical

---

**Algorithm 1** Stochastic Conditional Gradient (SCG)
---
**Input:** $x_0, m_0 \in \mathcal{X}$, parameters $\alpha, \beta \in (0,1), \eta > 0$
**for** $k = 0, \ldots, K-1$ **do**
    sample $\xi_k \sim \mathcal{D}$
    compute $m_{k+1} = (1-\alpha)m_k + \alpha g(x_k; \xi_k)$
    compute $d_{k+1} = \arg\min_{d \in \mathcal{X}} \langle m_{k+1}, d \rangle$ s.t. $\|d\| \leq 1$
    compute $x_{k+1} = (1-\beta)x_k + \beta\eta d_{k+1}$
**end for**

---

power-law relations between compute budget, batch size, and learning rate that perform well at scale.

Taken together, these works reinforce a central practical message: the best batch size is typically not a fixed constant, but depends on training stage, optimization hyperparameters, and budget. Motivated by this, we seek *principled, token-budget–aware* rules that characterize how the optimal effective batch–sequence scale and stepsize should co-vary with $T$, and how $(B, S, \beta)$ should be adapted.

## 3. Problem Formulation and Assumptions

We consider the following problem template:

$$\min_{x \in \mathcal{X}} f(x), \tag{1}$$

where the space $\mathcal{X}$ is equipped with a standard Euclidean norm $\|\cdot\|_2$ induced by the inner product $\langle\cdot,\cdot\rangle$, i.e., $\|x\|_2 = \sqrt{\langle x, x \rangle}$, and another norm $\|\cdot\|$, which possibly does not coincide with the Euclidean one. For the norm $\|\cdot\|$, we define the associated dual norm $\|x\|_* := \sup_{\|x'\| \leq 1} \langle x, x' \rangle$ for all $x \in \mathcal{X}$. We seek to solve (1) using Algorithm 1.

**Assumption 1.** *Let the gradient $\nabla f(\cdot)$ be Lipschitz continuous with respect to the norm $\|\cdot\|$:*

$$\|\nabla f(x) - \nabla f(x')\|_* \leq L\|x - x'\| \quad \forall x, x' \in \mathcal{X}, \quad \text{(A1)}$$

*where $L > 0$ is the gradient Lipschitz constant.*

**Assumption 2.** *There exist a constant $\rho > 0$ such that*

$$\|x\|_* \leq \rho\|x\|_2 \quad \forall x \in \mathcal{X}. \tag{A2}$$

*Note that such a constant always exists by norm equivalence, which always holds in finite-dimensional spaces $\mathcal{X}$.*

**Assumption 3.** *The objective function $f(x)$ is $\mu$-KL for some $\mu > 0$:*

$$\|\nabla f(x)\|_* \geq \mu(f(x) - f^\star) \quad \forall x \in \mathcal{X}, \tag{A3}$$

*where $f^\star = \min_{x \in \mathcal{X}} f(x)$.*

Note that condition (A3) is closely related to the Polyak-Łojasiewicz (PL) condition $\|\nabla f(x)\|_2^2 \geq \mu(f(x) - f^\star)$ originally studied in Polyak (1963); Łojasiewicz (1964). Variants of the PL condition have been investigated for over-parameterized models (Liu et al., 2022). A key distinction between the $\mu$-PL and $\mu$-KL conditions lies in the

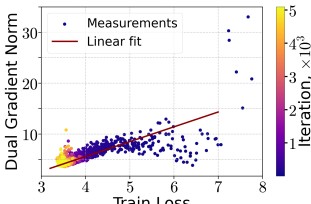

*Figure 1.* Empirical evidence for Assumption 3 during training of a 124M NanoGPT model. Points with training loss below 5 are fitted with Huber regression; the fitted slope provides the estimate of $\mu$, while the intercept absorbs the unknown shift involving $f^\star$.

exponent of the gradient norm, making the difference between them significant when the norm is small.

Nevertheless, condition (A3) has been extensively used in the optimization literature to analyze gradient descent under the Euclidean norm (Bolte et al., 2014; Fatkhullin et al., 2022). For problems with a bounded domain, the $\mu$-KL condition is closely related to $\zeta$-quasar convexity ($\zeta$-QC) (Hardt et al., 2018; Guminov et al., 2023), which requires $\langle \nabla f(x), x - x^\star \rangle \geq \zeta(f(x) - f^\star)$ for some $x^\star \in \mathcal{X}$ and all $x \in \mathcal{X}$. $\zeta$-QC naturally arises in the training of neural networks (Zhou et al., 2019; Kleinberg et al., 2018). When $\mathcal{X}$ is bounded with diameter $R$ with respect to the norm $\|\cdot\|$, $\zeta$-QC implies the $\mu$-KL condition with $\mu = \zeta/R$.

In this work, we extend the applicability of the standard $\mu$-KL assumption beyond the Euclidean norm. To provide empirical evidence for this condition in practice, we track the training loss and dual gradient norm during the training of a 124M NanoGPT model. In Figure 1, we observe that the measurements fit a linear function well, especially when the loss is below 5 (*cf.*, the description of the full setting in Section 6.1)

We make the assumption below for the gradient noise.

**Assumption 4.** *We have access to the unbiased estimator $g(\cdot; \xi): \mathcal{X} \to \mathcal{X}$ of the gradient $\nabla f(\cdot)$, where $\xi \sim \mathcal{D}$ is a random variable sampled from a probability distribution $\mathcal{D}$. We assume that the stochastic gradient estimator $g(\cdot; \xi)$ is unbiased and has $\sigma$-bounded variance for some $\sigma \geq 0$:*

$$\mathbb{E}_{\xi \sim \mathcal{D}}[g(x; \xi)] = \nabla f(x), \quad and \qquad (A4)$$
$$\mathbb{E}_{\xi \sim \mathcal{D}}[\|g(x; \xi) - \nabla f(x)\|_2^2] \leq \sigma^2.$$

*Additionally, let $\sigma^2 = \frac{\sigma_\star^2}{BS}$, where $B$ and $S$ are batch size and sequence length respectively.*

Assumption 4 is a classical assumption for the in-expectation convergence analysis of stochastic methods (Ghadimi & Lan, 2012; 2013). We provide empirical evidence for Assumption 4 during the training in Figure 2 (*cf.*, the description of the full setting in Section 6.1).

## 4. Theoretical Analysis

This section establishes convergence guarantees for Algorithm 1, guiding how to choose the batch size $B$, sequence length $S$, and stepsize $\beta$ under a fixed token budget $T$. The proof and the full statement of the following theorem are deferred to Section E.

**Theorem 1** (Simplified). *Let Assumptions* (A1), (A2), (A3), *and* (A4) *hold. Let $m_0 = g(x_0; \xi_0)$, the parameters of Algorithm 1 and initialization $x_0$ be chosen as follows*

$$\beta = \mathcal{O}\left(\frac{1}{K}\right), \quad \eta = \widetilde{\mathcal{O}}\left(\frac{1}{\mu}\right), \qquad (2)$$

$$\alpha = \min\left\{1, \mathcal{O}\left(\frac{(\varepsilon\mu)^2}{(\rho\sigma)^2}\right)\right\}, \quad 2\|x_0\| \leq \eta, \quad and$$

$$K = \max\left[\widetilde{\mathcal{O}}(1), \widetilde{\mathcal{O}}\left(\max\left\{\frac{L}{\varepsilon\mu^2}, \frac{\rho\sigma}{\varepsilon\mu}, \frac{L(\rho\sigma)^2}{\mu(\varepsilon\mu)^3}, \frac{(\rho\sigma)^3}{(\varepsilon\mu)^3}\right\}\right)\right],$$

*where $\mathcal{O}$ hides all numerical constants and $\tilde{\mathcal{O}}$ hides all numerical and logarithmic factors. Then, the output of Algorithm 1 after $K$ iterations satisfies $\mathbb{E}[f(x_K) - f^\star] \leq \varepsilon$.*

**Remark 1.** Convergence bounds for SCG were derived in Pethick et al. (2025a) for the Frank–Wolfe gap, then similar results to Theorem 1 were given by Kovalev (2025)[1] under star-convexity, a special case of $\zeta$-quasar convexity with $\zeta = 1$. In light of the relationship between the $\mu$-KL condition and $\zeta$-QC in Section 3, this similarity is expected.

Our work goes beyond this connection in two important ways. First, we provide empirical justification for the use of the $\mu$-KL condition in the analysis. Second, building on this framework, we derive new theory-guided scaling rules for both the learning rate and the batch size.

In practice, the number of iterations $K$ cannot be arbitrarily large. In fact, $K$ is trivially constrained by the available token budget $T$, the two being related by the simple identity $T = K \cdot B \cdot S$. Consequently, the requirement on $K$ in Theorem 1[2] can be equivalently expressed as a condition on $T$ by multiplying both sides by $BS$:

$$T = \tilde{\mathcal{O}}\left(\max\left\{\frac{LBS}{\varepsilon\mu^2}, \frac{\rho\sigma BS}{\varepsilon\mu}, \frac{L(\rho\sigma)^2 BS}{\mu(\varepsilon\mu)^3}, \frac{(\rho\sigma)^3 BS}{(\varepsilon\mu)^3}\right\}\right)$$

Under a fixed token budget, the expression above indicates that we cannot achieve an arbitrary optimization error $\varepsilon$. Instead, Corollary 1 gives the optimization-error guarantee obtained from Theorem 1 under a fixed token budget.

**Corollary 1** (BST Scaling Rule). Under the setup of Theorem 1, running the algorithm with parameters from Theo-

---

[1]Kovalev (2025) studies a stochastic first-order non-Euclidean trust-region method with momentum and weight decay, which is equivalent to Algorithm 1.

[2]We ignore the requirement $K = \widetilde{\mathcal{O}}(1)$, as it is always satisfied in practice; see also Corollary 2 for the details. In the sequel, we omit numerical constants for clarity.

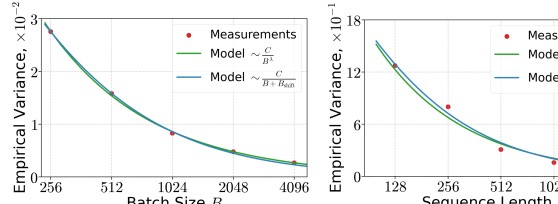

*Figure 2.* Empirical gradient variance and fitted power-law models as functions of batch size $B$ with fixed sequence length $S = 1024$ (**left**) and sequence length $S$ with fixed batch size $B = 512$ (**right**) when training a 124M NanoGPT model on the FineWeb dataset under a fixed token budget $T = 2.7B$. For the left plot, the estimated scaling exponent is $\lambda \approx 0.9$ and $B_{\text{shift}} \approx 90$, while for the right plot they are $\lambda \approx 1.1$ and $S_{\text{shift}} \approx 35$. The fitted models provide evidence for Assumption 4.

rem 1 for $\frac{T}{BS}$ iterations, we achieve the optimization error

$$\varepsilon = \tilde{\mathcal{O}}\left(\max\left\{\frac{LBS}{\mu^2 T}, \left(\frac{L\rho^2\sigma_\star^2}{\mu^4 T}\right)^{1/3}, \frac{\rho\sigma_\star}{\mu(T^2 BS)^{1/6}}\right\}\right). \quad (3)$$

Corollary 1 provides key insights into how the error $\varepsilon$ typically varies as the product $BS$ changes:

1. For small effective batch sizes, the third term in (3) dominates, and $\varepsilon$ improves as $BS$ increases.

2. When the effective batch size exceeds $\left(\frac{\mu\rho\sigma_\star}{L}\right)^2$, the second term in (3) becomes dominant. In this regime, the error is independent of the batch size and sequence length, and the error instead scales as $\sim T^{-1/3}$.

3. Further increasing $BS$ moves the system into an iteration-starved regime where the first term dominates, causing the error to deteriorate linearly in $BS$.

Corollary 1 indicates that the optimal achievable performance lies in the second regime, where the optimization error $\varepsilon$ is independent of both the batch size and the sequence length. From a practical perspective, however, larger batch sizes are often preferred to improve GPU utilization (Narayanan et al., 2021).

This motivates us to select the batch size and sequence length at the crossover between the second and third regimes. Following this intuition, we choose $B$ and $S$ as

$$\frac{L}{\mu^2}\frac{BS}{T} = \left(\frac{L\rho^2\sigma_\star^2}{\mu^4 T}\right)^{1/3} \Leftrightarrow BS = \left(\frac{T\mu\rho\sigma_\star}{L}\right)^{2/3}, \quad (4)$$

balancing final performance and hardware efficiency. Next, the BST rule results in the Frank–Wolfe stepsize

$$\beta_\star \sim 1/K. \quad (5)$$

Notably, a Frank–Wolfe stepsize of this form is used in practice when employing decoupled weight decay (Loshchilov & Hutter, 2019) to train LLMs near the Chinchilla-optimal token-per-parameter (TPP) regime

(Xiao, 2024; Qiu et al., 2025), where the model depth scales proportionally with the token budget.

Using $\varepsilon = \left(\frac{L\rho^2\sigma_\star^2}{\mu^4 T}\right)^{1/3}, BS = \left(\frac{T\mu\rho\sigma_\star}{L}\right)^{2/3}$, and Assumption 4 in Theorem 1, we obtain that the momentum parameter $\alpha$

$$\alpha \sim \frac{\mu^2 BS}{\rho^2\sigma_\star^2} \cdot \left(\frac{L\rho^2\sigma_\star^2}{\mu^4 T}\right)^{2/3} = \text{Const.}$$

This suggests that if we find an optimal momentum parameter $\alpha$ for a small model, under the BST scaling rule, it transfers to the larger setting.

To summarize, the BST scaling rule suggests the following choice of parameters in Algorithm 1:

$$BS \sim T^{2/3}, \quad \beta \sim \frac{1}{K}, \quad \alpha = \text{Const.} \quad (6)$$

Our theoretically derived BST rule is closely aligned with empirically obtained ones (Filatov et al., 2025). In Section 5, we provide a more detailed explanation of the BST scaling rule for the hyperparameter transfer.

## 5. Strategies for Hyperparameter Choice

**Training Setup.** Training a model such that (4) holds establishes working strategies on how to train a larger model of size $D_1$ efficiently, given that we have a tuned configuration (i.e., the tuned values of Frank–Wolfe stepsize $\beta_0$, momentum parameter $\alpha$, batch size $B_0$, and sequence length $S_0$) for a smaller model of size $D_0$. We consider the training under a fixed TPP, which implies that the available token budget increases proportionally to the model size, i.e., $T_1/T_0 = D_1/D_0$. Moreover, we assume that the problem constants $L = L(D), \mu = \mu(D)$, and $\rho = \rho(D)$ change with model size. We denote the constants with subscripts 1 and 0 for models of size $D_1$ and $D_0$, respectively.

**Remark 2.** In this work, we assume that the variance constant $\sigma_\star^2$ in Assumption 4 does not depend on the model size, as estimating its scaling with model size is computationally infeasible. We acknowledge, however, that in practice $\sigma_\star^2$ may change as the model size grows.

Given the training setup described above, we translate the results in (4) and (5) into actionable transfer rules for practical use. These rules are summarized in Table 1, with the derivations deferred to Section A. We consider two regimes: increasing the token budget while keeping the model size fixed, and simultaneously increasing both the token budget and the model size. Importantly, the BST scaling rule prescribes scaling the Frank–Wolfe stepsize $\beta$ as $T^{-1/3}$ and the batch size $BS$ as $T^{2/3}$. This highlights the need to adjust hyperparameters when transferring from a smaller token budget used for tuning to a larger token

*Table 1.* Summary of BST scaling rules for transferring hyperparameters to larger token budgets or model sizes. We explicitly separate the scaling with respect to the token budget $T$ from changes in the problem-dependent constants, which are captured by $\chi_{0\to1}$.

| Regime | Tuned at $T_0$ | | | Transfer Rule to $T_1$ | | | $\chi_{0\to1}$ |
| --- | --- | --- | --- | --- | --- | --- | --- |
| | FW Stepsize | Batch Size × Sequence Length | Momentum | FW Stepsize | Batch Size × Sequence Length | Momentum | |
| Increasing Token Budget | $\beta_0^\star$ | $B_0^\star S_0^\star$ | $\alpha$ | $\beta_1 = \beta_0^\star\left(\frac{T_1}{T_0}\right)^{-1/3}\chi_{0\to1}^{2/3}$ | $B_1 S_1 = B_0^\star S_0^\star\left(\frac{T_1}{T_0}\right)^{2/3}\chi_{0\to1}^{2/3}$ | $\alpha$ | $\frac{\rho_1}{\rho_0}$ |
| Increasing Token Budget and Model Size | | | | | | | $\frac{\mu_1\rho_1 L_0}{\mu_0\rho_0 L_1}$ |

budget or model for which direct tuning is infeasible or prohibitively expensive.

In the following, we propose a principled and practically implementable pipeline for selecting and adapting batch size and sequence length in the delayed-data regime.

**First stage (training with $T_{(1)} = T_0$ tokens).** Assume that in the beginning, we only have a smaller token budget $T_{(1)} = T_0$, which is sufficient to train a smaller model efficiently, but insufficient to do so for a larger model. The remaining tokens $T_{(2)} = T_1 - T_0$ arrive at a later time. Based on Table 1 (i.e., (9) and (10)), when training the large model using $T_{(1)}$ tokens,[3] our theory suggests choosing the batch size $B_1$, sequence length $S_1$, and Frank–Wolfe stepsize such that

$$B_1 S_1 = B_0^\star S_0^\star \chi_{0\to1}^{2/3} \overset{(a)}{\approx} B_0^\star S_0^\star,$$
$$\beta_1 = \beta_0^\star \chi_{0\to1}^{2/3} \overset{(b)}{\approx} \beta_0^\star, \tag{7}$$

where (a) and (b) holds if problem constants do not change significantly, that is, the effective batch–sequence scale for the large model should closely match that of the small model when $\chi_{0\to1} \approx 1$ and the problem-dependent constants do not vary substantially with the model size (see Section 6.3). Such a choice of the Frank–Wolfe stepsize is also recommended by the $\mu$P literature, which advocates keeping the learning rate fixed when the token budget and batch configuration are unchanged.

**Second stage (training with the full budget $T_{(1)} + T_{(2)}$).** Next, we receive an additional $T_{(2)}$ tokens. Eq. 3 suggests that we should expect the optimization error to improve from order $T_0^{-1/3}$ at the end of the first stage to order $T_1^{-1/3}$ at the end of the second stage. To realize this improvement in practice, we switch to using hyperparameters from Table 1 (i.e., (9) and (10)) during the second stage, with the full token budget $T_1 = T_{(1)} + T_{(2)}$.

Overall, this hyperparameter restart strategy for Scion suggests selecting the batch size, sequence length, and Frank–Wolfe stepsize based on the total number of tokens that will ultimately be available to the model. If additional tokens arrive at later times, the same procedure can be repeated: the batch size and sequence length are increased accordingly,

---

[3]We should use $T_{(1)}$ instead of $T_1$ in (9) and (10) or Table 1.

and the Frank–Wolfe stepsize is adjusted based on the final token budget that the larger model will observe.

## 6. Experiments

In this section, we empirically evaluate our theoretical results by training a modded NanoGPT model on the FineWeb dataset, following the experimental setup of Pethick et al. (2025a) and based on the codebase of Jordan et al. (2024a). Details are given in Section B. For Scion, we adopt the recommended operator norms (Sign → Spectral → Sign): we choose the radius $\eta = 3000$ for sign-updated layers and $\eta = 50$ for matrix-type layers. This corresponds to using the polar factor of the gradient for matrix-valued parameters and the elementwise sign of the gradient for all other parameter types (*cf.*, (Pethick et al., 2025a)).

### 6.1. Verification of Assumptions 4 and 3

First, we empirically test the validity of Assumption 4 when training a 124M base model with Scion for a fixed number of iterations $K = 5100$. To approximate the gradient variance as a function of the batch size $B$, we sample $m$ mini-batch gradients of size $B$ such that $mB = 32768$, and compute the empirical variance across the sampled $m$ mini-batch gradients. We track the evolution of this empirical variance over training in Figure B.1 and observe that it stabilizes rapidly after a short initial transient phase. In Figure 2, we report the final empirical variance values measured at the end of training. The fitted power-law relationships support $\sigma^2 \sim \frac{1}{BS}$ as a reasonable working approximation in the regime $BS \ll T$.

Second, we conduct experiments to assess the validity of Assumption 3 in practice. We use the same experimental setup as in the previous section and track both the dual norm of mini-batch gradients and the corresponding mini-batch training loss throughout training. When using Scion, the primal and dual norms are defined as

$$\|x\| = \max_{\ell\in[N]} \|x_\ell\|_\ell, \quad \|x\|_* = \sum_{\ell=1}^N \|x_\ell\|_{*,\ell},$$

where $\|x_\ell\|_\ell$ and $\|x_\ell\|_{*,\ell}$ denote the primal and dual norms of the $\ell$-th layer of the network with $N$ layers, respectively. Their precise definitions are provided in Table 2 (second and third columns) of Pethick et al. (2025a). See also the

*Table 2.* Final validation loss when training a 124M NanoGPT model under the token budget 1.3B (TPP 10.8), varying the batch size (**left**) and the train sequence length (**right**). **Left:** validation and train sequence lengths are fixed to 1024. **Right:** batch size is fixed to 256; validation sequence length is always 1024; [†] indicates that not all runs had a stable decrease in validation loss. In both tables, we report the average across 5 runs along with a standard deviation. **Bold** numbers indicate the best performance in the column. The runs in red indicate the best configuration of batch size, sequence length, and Frank–Wolfe stepsize across all runs for a given token budget.

| $\beta, \times 10^{-4}$ | Batch Size | | | | | | Sequence Length | | | | |
|---|---|---|---|---|---|---|---|---|---|---|---|
| | 64 | 128 | 256 | 512 | 1024 | 2048 | 256 | 512 | 1024 | 2048 | 4096 |
| 1.2 | $\mathbf{3.4258}_{\pm 0.0004}$ | $3.3889_{\pm 0.0012}$ | $3.3857_{\pm 0.0013}$ | $3.4074_{\pm 0.0010}$ | $3.4587_{\pm 0.0010}$ | $3.5598_{\pm 0.0012}$ | $3.7076_{\pm 0.0084}$ | $3.4647_{\pm 0.0240}$ | $3.4587_{\pm 0.0010}$ | $3.4126_{\pm 0.0014}$ | $3.4811_{\pm 0.0025}$ |
| 2.4 | $3.4394_{\pm 0.0007}$ | $\mathbf{3.3880}_{\pm 0.0043}$ | $3.3706_{\pm 0.0019}$ | $3.3801_{\pm 0.0016}$ | $3.4144_{\pm 0.0004}$ | $3.4940_{\pm 0.0009}$ | $3.9622_{\pm 0.0585}{}^{\dagger}$ | $\mathbf{3.4633}_{\pm 0.0091}$ | $3.3706_{\pm 0.0019}$ | $3.3834_{\pm 0.0026}$ | $3.4299_{\pm 0.0021}$ |
| 3.6 | $3.4554_{\pm 0.0008}$ | $3.3945_{\pm 0.0015}$ | $\mathbf{3.3717}_{\pm 0.0020}$ | $\mathbf{3.3765}_{\pm 0.0017}$ | $\mathbf{3.4065}_{\pm 0.0007}$ | $3.4799_{\pm 0.0017}$ | $4.0441_{\pm 0.2055}{}^{\dagger}$ | $3.4796_{\pm 0.0131}$ | $\mathbf{3.3717}_{\pm 0.0020}$ | $\mathbf{3.3792}_{\pm 0.0022}$ | $\mathbf{3.4216}_{\pm 0.0013}$ |
| 4.8 | $3.4766_{\pm 0.0016}$ | $3.4072_{\pm 0.0048}$ | $3.3790_{\pm 0.0013}$ | $3.3807_{\pm 0.0006}$ | $3.4115_{\pm 0.0025}$ | $3.4945_{\pm 0.0038}$ | $3.9292_{\pm 0.0852}{}^{\dagger}$ | $3.5004_{\pm 0.0063}$ | $3.3790_{\pm 0.0013}$ | $3.3829_{\pm 0.0020}$ | $3.4243_{\pm 0.0029}$ |
| 6.0 | $3.4967_{\pm 0.0022}$ | $3.4198_{\pm 0.0002}$ | $3.3875_{\pm 0.0019}$ | $3.3887_{\pm 0.0024}$ | $3.4202_{\pm 0.0022}$ | $3.5005_{\pm 0.0038}$ | $3.9901_{\pm 0.0170}{}^{\dagger}$ | $3.5134_{\pm 0.0059}$ | $3.3875_{\pm 0.0019}$ | $3.3910_{\pm 0.0024}$ | $3.4374_{\pm 0.0037}$ |
| 7.2 | $3.5151_{\pm 0.0007}$ | $3.4301_{\pm 0.0001}$ | $3.3978_{\pm 0.0026}$ | $3.3960_{\pm 0.0022}$ | $3.4331_{\pm 0.0025}$ | $3.5270_{\pm 0.0071}$ | $3.9819_{\pm 0.1195}{}^{\dagger}$ | $3.5269_{\pm 0.0187}$ | $3.3960_{\pm 0.0022}$ | $3.3987_{\pm 0.0029}$ | $3.4537_{\pm 0.0030}$ |

recent work by Crawshaw et al. (2026).

We report the joint evolution of the dual gradient norm and the training loss over the course of training in Figure 1. We observe that, once the training loss falls below approximately 5, the data points closely follow a linear relationship, empirically supporting the use of Assumption 3 in this setting. To quantify this relationship, we estimate the slope using a robust linear regression model with Huber loss, which interpolates between least squares and absolute-error ($\ell_1$) regression and thereby reduces sensitivity to outliers.

## 6.2. Ablations on Batch Size and Sequence Length

We conduct ablation studies by varying the batch size $B$ and sequence length $S$ to identify the optimal Frank–Wolfe stepsize $\beta$ for Scion when training a base 124M model with a fixed validation sequence length 1024. We report results under a fixed token budget of 1.3B in Table 2. This corresponds to the TPP ratio of 10.8 (approximately $0.5\times$ the Chinchilla optimum).

We observe that once the batch size or sequence length is sufficiently large, the optimal Frank–Wolfe stepsize stabilizes at $3.6 \cdot 10^{-4}$. Moreover, the results indicate that, for the base model, the optimal batch size and sequence length are approximately 256 and 1024, respectively, which yield the lowest validation loss. For substantially shorter training sequence lengths, such as 256, most runs were unstable and exhibited high standard deviations, likely because validation was performed at sequence length 1024. We also observe that the difference in performance between batch sizes 256 and 512 is small, suggesting that performance is nearly batch-independent in this regime, consistent with Corollary 1.

## 6.3. Estimating Problem-Dependent Constants

In our next experiment, we estimate the problem-dependent constants $L$, $\mu$, and $\rho$ across different model configurations in order to track how these quantities change with model size. Specifically, we train models using a fixed Frank–Wolfe stepsize $\beta = 3.6 \cdot 10^{-4}$, batch size $B = 512$, and sequence length $S = 1024$ for 5100 iterations, following the ablation study in Section 6.2, while varying the number of layers $n_l \coloneqq$ n_layer and the embedding dimension

*Table 3.* Estimated problem-dependent constants, assuming that they change with the number of layers n_layer, embedding dimension n_embd, and batch size according to (8). The estimations of the change for $\beta$ and $BS$ are based on (9) and (10).

| Model | $L$ | $\mu$ | $\rho$ | How to change $\beta$ w.r.t. 124M model? | How to change $BS$ w.r.t. 124M model? |
|---|---|---|---|---|---|
| **124M** | 7.2 | 3.1 | 62.7 | $1\times$ | $1\times$ |
| **1B** | 10.6 | 2.9 | 111.9 | $\searrow 0.5 \times$ [a] $\searrow 0.54 \times$ [b] | $\nearrow 4 \times$ [a] $\nearrow 4.37 \times$ [b] |

[a] Taking into account the practical requirement that $B$ and $S$ should be powers of two. We increase the product $BS$, rounding to the closest power of two.

[b] Ignoring the practical requirement that $B$ and $S$ should be powers of two.

$n_e \coloneqq$ n_embd. In this section, we ignore the change in the constants $L, \mu, \rho$ with the batch size, but later we account for this dependency in the hyperparameter transfer. The estimated values are reported in Section C. The estimation procedure is carried out as follows.

**Smoothness constant $L$.** To estimate the smoothness constant, we measure the following ratio

$$\frac{\|g(x_k; \xi_k) - g(x_{k-1}; \xi_{k-1})\|_*}{\|x_k - x_{k-1}\|},$$

where $g(x_k; \xi_k)$ and $g(x_{k-1}; \xi_{k-1})$ denote the mini-batch gradients at two consecutive iterations, and the norms are defined as in the previous section. This quantity has been used in prior work as a proxy for local curvature during training (Alimisis et al., 2026; Zhang & Sennrich, 2019; Riabinin et al., 2026). As a final estimate of $L$, we average the measured ratio over the last 100 iterations.

**KL condition constant $\mu$.** The estimation of $\mu$ follows the same procedure as in Section 6.1. In particular, we fit a robust linear regression model with Huber loss to the relationship between the dual gradient norm and the training loss, and use the resulting slope as an estimate of $\mu$.

**Norm-equivalence constant $\rho$.** In the proof of Theorem 1, we apply Assumption 2 to bound terms of the form

$$\|g(x_k; \xi_k) - \nabla f(x_k)\|_* \le \rho \|g(x_k; \xi_k) - \nabla f(x_k)\|_2.$$

To approximate the full gradient $\nabla f(x_k)$, we follow the same procedure described in Section 6.1. We track the ratio between the dual norm and the Euclidean norm throughout training, and report the average of this ratio over the last 100 iterations as an estimate of $\rho$.

We conduct the estimation procedure for several model

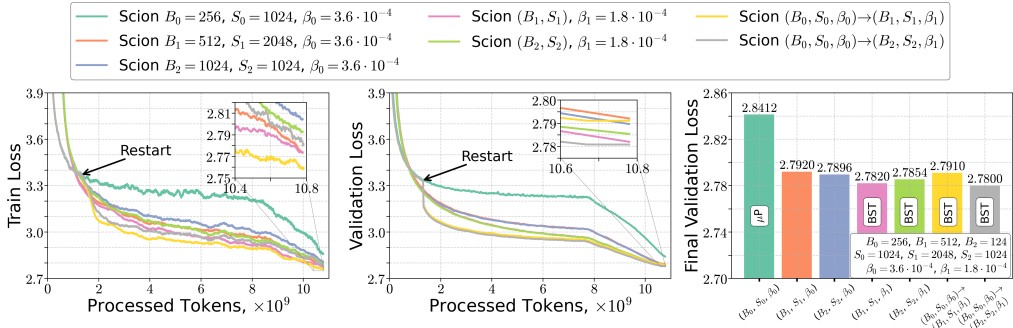

*Figure 3.* Batch-size and sequence-length scheduling strategies for the 1B model. Left/middle: smoothed train and validation loss versus processed tokens; right: final validation loss. Yellow and gray curves denote restarted Scion runs, while the other curves denote fixed tuned-batch and fixed large-$BS$ baselines trained from scratch. The legend reports the tuples $(B, S, \beta)$; arrows indicate the restart from $(B_0, S_0, \beta_0)$ to the corresponding $(B_1, S_1, \beta_1)$. The labels $\mu$P and BST indicate the rule used to select $B, S, \beta$.

configurations and fit a shifted power law[4] for the problem constants $L, \mu, \rho$ of the form:

$$\mu(n_l, n_e) = 5.2(n_l + 1.7)^{-0.2}; \qquad (8)$$

$$L(n_l, n_e) = 0.4(n_l + 0.7)^{0.2}(n_e + 126)^{0.35};$$

$$\rho(n_l, n_e, B) = 4.1(n_l - 2.7)^{0.25}(n_e - 250.8)^{0.3}(B - 9.4)^{0.1}.$$

Interestingly, the constant $\mu$ decreases with `n_layer`, while it remains unchanged with `n_embd`. In contrast, the constants $L$ and $\rho$ increase with both `n_layer` and `n_embd`. Using the fitted power laws, we estimate the constants for 2 model configurations used in the experiments of size 124M and 1B.

We observe that the problem-dependent constants vary slowly with model size and remain relatively stable across the model configurations we consider. Although we use these estimates in subsequent experiments, neglecting this variation does not significantly affect the resulting derivations. The impact of these changes becomes more pronounced only in regimes where $D_1 \gg D_0$. Using estimated constants, we characterize how the Frank–Wolfe stepsize $\beta$ and the product $BS$ should be set for the 1B model, knowing the optimal configuration ($B = 256, S = 1024, \beta = 3.6 \cdot 10^{-4}$) for the 124M model in Table 3 and using (9), (10). Note that we provide two configurations for the 1B model: whether the practical requirement that the batch size and sequence length be powers of 2 should be taken into account.

### 6.4. Increasing Batch Size and Sequence Length during Training

Next, we evaluate the proposed strategy from Section 5. We use the 124M model as a base model, for which we previously identified batch size $B_0 = 256$, sequence length $S_0 = 1024$, and Frank–Wolfe stepsize $\beta_0 = 3.6 \cdot 10^{-4}$ as providing the best performance under a token budget $T_0 =$

1.3B. We then consider training a larger 1B model under a total token budget of $T_1 = 10.8$B (the same TPP) using the following strategies.

We compare against three groups of baselines, whose hyperparameters are summarized in Table B.2 and described in full in Section B.1[5]. *Restarted Scion* trains with a small batch size $B_0 = 256$, $S_0 = 1024$ for the first $T_0$ tokens, then restarts with a larger batch-size–sequence-length product for the remaining $T_1 - T_0$ tokens taken from Table 1, following the strategy described in Section 5. *Fixed tuned-batch Scion* trains over the full horizon $T_1$ using the hyperparameters tuned on a smaller 124M model ($B_0 = 256$, $S_0 = 1024$, $\beta_0 = 3.6 \cdot 10^{-4}$), motivated by $\mu$P transfer. *Fixed large-batch Scion* trains from scratch over the full $T_1$ tokens with a larger batch-size–sequence-length product taken from Table 1, using either the $\mu$P-transferred stepsize $\beta_0$ or the rescaled stepsize $\beta_1$ from Table 1.

Based on the results in Figure 3, we can make the following claims.

1. The $\mu$P framework, where all parameters of the algorithm remain unchanged, achieves the worst performance. This result illustrates a regime not addressed by $\mu$P-style transfer, which keeps $B, S, \beta$ fixed and therefore does not account for token-budget-dependent changes in the effective scale $BS$. Our BST scaling instead suggests increasing the product $BS$ that leads to enhanced performance.

2. Both restarting strategies for Scion demonstrate competitive performance compared to the other baselines. After the restart, both variants show accelerated improvement in training and validation loss relative to the baselines. The training curves of the restarted

---

[4]The choice of the fitting model is flexible, and alternative functional forms could also be considered. We leave the exploration of other functional dependencies to future work.

[5]We note that the choice of batch size $B$ and sequence length $S$ in this set of experiments is partially guided by practical considerations, as these values are typically selected as powers of 2. We follow this convention to evaluate the performance of the restarted, small- and large-batch baselines in a setting that more closely reflects real-world practice.

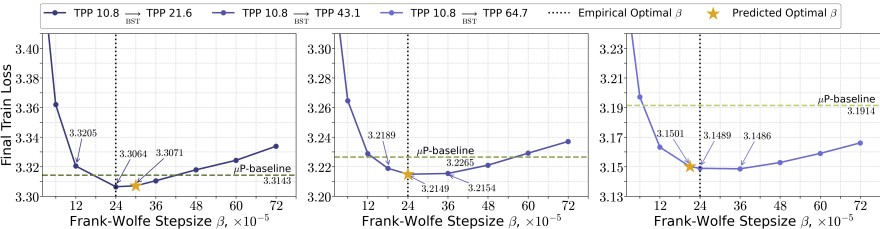

*Figure 4.* Final smoothed train loss for 124M model when varying the Frank–Wolfe stepsize $\beta$ under different token budgets (**left:** 2.7B, **center:** 5.3B, **right:** 8.0B). We observe that the BST scaling rule predicts a good estimate for the optimal $\beta$ when increasing the token budget. Moreover, the difference in performance between BST and $\mu$P baselines grows with a token budget.

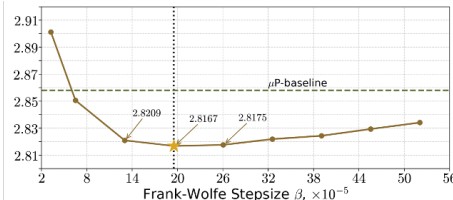

*Figure 5.* Final smoothed train loss for the 1B model when varying Frank–Wolfe stepsize $\beta$ under a token budget of 10.8B tokens (TPP 10.8). We observe that the BST scaling rule predicts a good estimate for optimal $\beta$ when transferring from a smaller 124M model to a larger 1B model.

Scion models remain consistently below those of the other methods from the restart point onward.

3. Among the restarted schedules, increasing $B$ from 256 to 1024 while keeping $S = 1024$ slightly outperforms the restart that increases both $B$ and $S$ to $B = 512, S = 2048$. In this setting, the former strategy achieves approximately a 0.01 lower validation loss than the latter.

4. All large-batch baselines (with both values of Frank–Wolfe stepsize: $\beta_0$ suggested by $\mu$P and $\beta_1$ suggested by BST rule) achieve similar performance. Among the fixed large-$BS$ baselines trained from scratch, the $B = 512, S = 2048$ configuration with the BST-rescaled Frank–Wolfe stepsize $\beta_1$ achieves the best validation loss. The other fixed large-BS baselines are slightly worse, with validation losses approximately 0.005–0.01 higher.

### 6.5. Frank–Wolfe Stepsize Transfer

Having the same tuned 124M model ($B = 256, S = 1024, \beta = 3.6 \cdot 10^{-4}$ under $T = 1.3$B token budget), we test the BST rule when transferring the parameters to larger training horizon or model size. We report smoothed train losses over a window of size 500.

**Increasing Token Budget.** In this section, we report pretraining results for the 124M model under increased token budgets: $(i)$ $T = 2.7$B (TPP 21.6), $(ii)$ $T = 5.3$B (TPP 43.1), and $(iii)$ $T = 8.0$B (TPP 64.7). We set the batch sizes for these longer horizons according to Table 1: $B = 416$ for $T = 2.7$B, $B = 672$ for $T = 5.3$B, and $B = 896$ for $T = 8.0$B. These values are obtained us-

ing estimates of the problem-dependent constants from Table 3, while fixing $\alpha = 0.1$ and $S = 1024$.

To demonstrate the predictiveness of the BST scaling rule in finding optimal Frank–Wolfe stepsize, we report the final train losses when varying $\beta$ under three token budgets. We expect the optimal Frank–Wolfe stepsize to be around $(i)$ $\beta = 3.0 \cdot 10^{-4}$, $(ii)$ $\beta = 2.4 \cdot 10^{-4}$, and $(iii)$ $\beta = 2.1 \cdot 10^{-4}$. In Figure 4, we observe that the BST scaling rule predicts Frank–Wolfe stepsize close to the optimal one in all the cases. Moreover, we observe that $\mu$P baseline ($B = 256, S = 1024, \beta = 3.6 \cdot 10^{-4}$) becomes more suboptimal when increasing the token budget, which shows the limitations of the $\mu$P framework even further.

**Increasing Model Size.** Now we want to train a 1B model ($B = 1120, S = 1024, T = 10.8$B). In this setup, we test the predictive power of the BST scaling rule when we change the model size. The value of the batch size is set according to Table 1, using estimates from Table 3. We expect the optimal Frank–Wolfe stepsize to be close to $1.95 \cdot 10^{-4}$. We report the results in Figure 5. We observe that the BST rule provides a good estimate of the optimal Frank–Wolfe stepsize $\beta$ when increasing the model size.

**Additional Experiments.** In Section D, we present additional experiments and ablation studies that provide further evidence for the strong predictive power of the BST rule.

## 7. Conclusion

We developed a token-budget–aware theory for scaling batch size, sequence length, and stepsize in SCG methods under a $\mu$-KL condition. Our analysis reveals a non-monotone dependence of optimization error on the effective batch–sequence scale and yields a principled BST-scaling rule that identifies when increasing batch size is beneficial and when it becomes suboptimal. In contrast to hyperparameter transfer approaches that ensure local stability at initialization, our results characterize long-horizon, trajectory-level behavior and explain how HPs should adapt as the token budget grows. Empirical results confirm that large batches are not inherently harmful: when scaled according to theory, jointly adapting $(B, S, \beta)$ improves token efficiency and convergence in large-scale training.

# Acknowledgement

Rustem Islamov and Aurelien Lucchi acknowledge the financial support of the Swiss National Science Foundation, SNSF grant No 207392. Volkan Cevher acknowledges the financial support of the Swiss National Science Foundation, SNSF grant No 240094. This work was also supported under project ID # 37 as part of the Swiss AI Initiative, through a grant from the ETH Domain and computational resources provided by the Swiss National Supercomputing Centre (CSCS) under the Alps infrastructure.

# Impact Statement

This work studies optimization rules for choosing batch size, sequence length, and stepsize in stochastic conditional-gradient training. Its most direct potential benefit is improved token and hardware efficiency, which may reduce wasted compute during model development; we do not introduce new datasets, deployed systems, or capabilities that create distinct societal risks beyond those associated with large-scale model training.

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

# A. Guidelines on How to Use BST Rule

## A.1. Increasing Batch Size

We assume that the optimal batch size $B_0^\star$, sequence length $S_0^\star$, and $\beta_0^\star$ are tuned for a small model[6] of size $D_0$ and satisfy (4), namely

$$B_0^\star S_0^\star \sim \left( \frac{T_0 \mu_0 \rho_0 \sigma_\star}{L_0} \right)^{2/3}.$$

We now determine $B_1$ and $S_1$ such that (4) remains satisfied for a larger model. By defining $\chi_{0\to1} := \frac{\mu_1}{\mu_0} \frac{\rho_1}{\rho_0} \frac{L_0}{L_1}$, a simple manipulation gives

$$B_1 S_1 = B_0^\star S_0^\star \left( \frac{T_1}{T_0} \chi_{0\to1} \right)^{2/3}. \tag{9}$$

Note that the ratio $T_1/T_0$ can be replaced by $D_1/D_0$ under fixed TPP. Knowing how $L, \mu, \rho$ change with model size and batch size,[7] we can adjust the batch size and sequence length for a larger model.

## A.2. Tuning the Frank–Wolfe Stepsize

From (5) we know that the optimal Frank–Wolfe stepsize $\beta$ should scale as $\frac{1}{K}$; therefore, we have

$$\frac{\beta_0^\star}{\beta_1} = \frac{B_0^\star S_0^\star / T_0}{B_1 S_1 / T_1} \Rightarrow \beta_1 = \beta_0^\star \frac{B_1 S_1}{B_0^\star S_0^\star} \frac{T_0}{T_1}. \tag{10}$$

Since we increase batch size and sequence length according to (9), then the optimal Frank–Wolfe stepsize for a larger model is expected to be around

$$\beta_1 = \beta_0^\star \left( \frac{T_1}{T_0} \right)^{-1/3} \chi_{0\to1}^{2/3}. \tag{11}$$

## A.3. Guidelines for Practitioners

We summarize all the details on how to adjust the optimizer's parameters under the BST scaling rule below to facilitate its implementation in practice.

### A.3.1. HYPERPARAMETER SCALING: FROM SMALL TO LARGE MODELS

In this scenario, the model size changes. Therefore, we need to account for a change of optimization problem constants, such as $L, \mu, \rho$. We summarize the resulting procedure below:

1. Obtain optimal values of the batch size $B_0^\star$ and sequence length $S_0^\star$, Frank–Wolfe stepsize $\beta_0^\star$ by tuning a small model, while setting momentum parameter $\alpha$ and radii $\eta$ to default values.

2. Estimate the problem constants $L_0, \mu_0, \rho_0$ and $L_1, \mu_1, \rho_1$ for small and large models, respectively, based on the fitted power laws (8).

3. Choose batch size $B_1$, sequence length $S_1$, and Frank–Wolfe stepsize $\beta_1$ for larger model using (9) and (10), namely

$$B_1 S_1 = B_0^\star S_0^\star \left( \frac{\frac{T_1}{T_0} \frac{\mu_1}{\mu_0} \frac{\rho_1}{\rho_0}}{\frac{L_1}{L_0}} \right)^{2/3}, \quad \beta_1 = \beta_0^\star \left( \frac{\frac{\sqrt{T_0}}{\sqrt{T_1}} \frac{\mu_1}{\mu_0} \frac{\rho_1}{\rho_0}}{\frac{L_1}{L_0}} \right)^{2/3}, \tag{12}$$

while keeping radii $\eta$ and momentum $\alpha$ unchanged.

---

[6] Ideally, we want all hyperparameters of the optimizer and model to be tuned for a small model, including radii $\eta$ or the initialization. However, such a task is infeasible even for a small model. Therefore, we focus on the main hyperparameters that affect the final performance the most: batch size, sequence length, and Frank–Wolfe stepsize, while we set the rest according to default values obtained from prior work.

[7] In real-world applications, the change of constants with a model size might be ignored for simplicity ($\chi = 1$), but later we provide estimates for them that we use in Section 6.

*Table B.1.* The model configurations and training details used in Section 6.

| Hyperparameter | 124M Model | 775M Model | 1B Model |
|---|---|---|---|
| Layers | 12 | 36 | 18 |
| Heads | 6 | 20 | 16 |
| Embedding Size | 768 | 1280 | 2048 |
| Weight Tying | | Yes | |
| Activation Function | | ReLU$^2$ | |
| Vocabulary Size | | 50304 | |
| Dataset | | FineWeb | |
| Warmdown | | 28% of the total token budget | |
| Stepsize Schedule | | $\beta_k = \begin{cases} \gamma & \text{if } k < n - m \\ \gamma \cdot \frac{n-k}{m} & \text{otherwise} \end{cases}$ | |
| Gradient Clipping | | No | |
| Momentum Parameter | | $\alpha = 0.1$ | |
| `lm_head/embd` Radii | | 3000 | |
| Matrix Weights Radii | | 50 | |
| Precision | | bf16 | |
| Device Batch Size | 32 *unless otherwise stated* | | 16 |

4. Use new parameters to train a larger model (either from the beginning or after processing the token budget used for tuning a smaller model).

### A.3.2. Hyperparameter Scaling: From Small to Large Token Budget

Now assume that the model size remains the same, but the token budget increases. Therefore, the constants $L$ and $\mu$ remain the same, while we need to account for a change of $\rho$ with batch size.

1. Obtain optimal values of the batch size $B_0$ and sequence length $S_0$, Frank–Wolfe stepsize $\beta_0$ by tuning a model for a smaller token budget, while setting momentum parameter $\alpha$ and radii $\eta$ to default values.

2. Estimate the problem constants $\rho_0$ and $\rho_1$ for small and large token budgets, respectively, based on the fitted power laws (8).

3. Choose batch size $B_1$, sequence length $S_1$, and Frank–Wolfe stepsize $\beta_1$ for a larger token budget $T_1$ using (9) and (10), namely

$$B_1 S_1 = B_0^\star S_0^\star \left( \frac{T_1}{T_0} \frac{\rho_1}{\rho_0} \right)^{2/3}, \quad \beta_1 = \beta_0^\star \left( \frac{\sqrt{T_0}}{\sqrt{T_1}} \frac{\rho_1}{\rho_0} \right)^{2/3}, \tag{13}$$

while keeping radii $\eta$ and momentum $\alpha$ unchanged.

4. Use new parameters to train a model for a longer horizon $T_1$ (either from the beginning or after processing the token budget used for a smaller model).

## B. Description of the Experimental Setup

Our implementation uses Scaled ReLU$^2$ from Large et al. (2024) (see Appendix B.2), rotary embeddings (Su et al., 2024) in place of positional embeddings, RMSNorm (Zhang & Sennrich, 2019) (without learnable parameters following Pethick et al. (2025a)) instead of LayerNorm, and a linear learning rate decay schedule instead of cosine annealing. The choice of radius is taken from Pethick et al. (2025a): $\eta = 50$ for matrix-type layers and $\eta = 3000$ for the rest of the layers. To approximate the polar factor of the gradient, we use the Newton-Schulz method with 5 iterations, following (Jordan et al., 2024b). All other details are reported in Table B.1.

Note that due to limited GPU availability, the 1B model is trained using checkpointing. This introduces slight fluctuations across runs. Although the random seed is fixed, some variability remains due to nondeterminism in the PyTorch implementation.

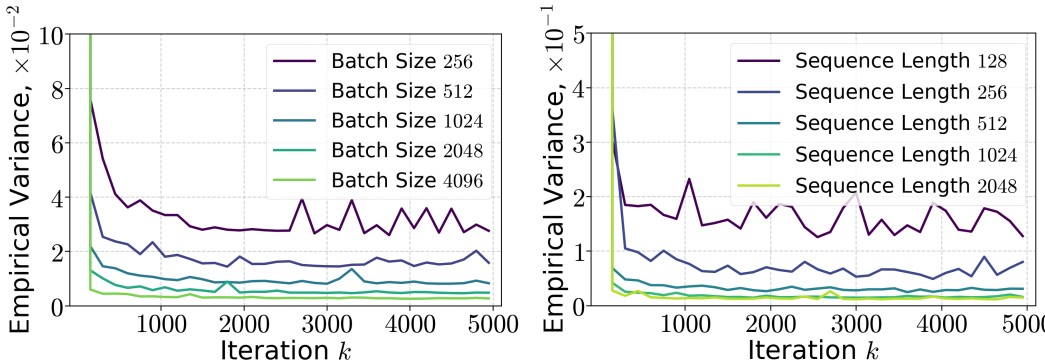

*Figure B.1.* Evolution of the empirical gradient variance varying batch size $B$ with fixed sequence length $S = 1024$ (**left**) and sequence length $S$ with fixed batch size $B = 512$ (**right**) when training a 124M NanoGPT model on the FineWeb dataset. We observe that the variance quickly stabilizes after a short initial phase.

*Table B.2.* Summary of baselines used in Section 6.4 and Figure 3. For each baseline, we report the batch size $B$, sequence length $S$, and Frank–Wolfe stepsize $\beta$ used during the first $T_0$ tokens and the remaining $T_1 - T_0$ tokens.

| Baseline | Color | $\mathbf{T_0}$ | | | $\mathbf{T_1 - T_0}$ | | |
|---|---|---|---|---|---|---|---|
| | | $B$ | $S$ | $\beta$ | $B$ | $S$ | $\beta$ |
| **Restarted Scion** | yellow | 256 | 1024 | $3.6 \cdot 10^{-4}$ | 512 | 2048 | $1.8 \cdot 10^{-4}$ |
| | gray | 256 | 1024 | $3.6 \cdot 10^{-4}$ | 1024 | 1024 | $1.8 \cdot 10^{-4}$ |
| **Fixed tuned-batch Scion** | light blue | 256 | 1024 | $3.6 \cdot 10^{-4}$ | 256 | 1024 | $3.6 \cdot 10^{-4}$ |
| **Fixed large-batch Scion** | orange | 512 | 2048 | $3.6 \cdot 10^{-4}$ | 512 | 2048 | $3.6 \cdot 10^{-4}$ |
| | pink | 512 | 2048 | $1.8 \cdot 10^{-4}$ | 512 | 2048 | $1.8 \cdot 10^{-4}$ |
| | blue | 1024 | 1024 | $3.6 \cdot 10^{-4}$ | 1024 | 1024 | $3.6 \cdot 10^{-4}$ |
| | green | 1024 | 1024 | $1.8 \cdot 10^{-4}$ | 1024 | 1024 | $1.8 \cdot 10^{-4}$ |

## B.1. Full Description of the Baselines in Section 6.4

We evaluate the proposed strategy from Section 5. We use the 124M model as a base model, for which we previously identified batch size $B_0 = 256$, sequence length $S_0 = 1024$, and Frank–Wolfe stepsize $\beta_0 = 3.6 \cdot 10^{-4}$ as providing the best performance under a token budget $T_0 = 1.3\text{B}$. We then consider training a larger 1B model under a total token budget of $T_1 = 10.8\text{B}$ (the same TPP) using the following strategies.

- **Restarted Scion.** We follow the batch size scheduling strategy described in Section 5. During the first stage, corresponding to the initial $T_0$ tokens, we use batch size $B_0 = 256$ and sequence length $S_0 = 1024$ with stepsize $\beta_0 = 3.6 \cdot 10^{-4}$. After processing $T_0$ tokens, we increase the product $BS$ four times and consider two restarted schemes: $B_1 = 512$ and sequence length $S_1 = 2048$ (in yellow) and $B_2 = 1024$ and sequence length $S_2 = 1024$ (in gray), both with stepsize $\beta_1 = \beta_0/2 = 1.8 \cdot 10^{-4}$ for the remaining token budget,[8] following the derivations in (9) and (10), with estimates of the problem-dependent constants taken from Section 6.3.

- **Fixed *tuned*-batch Scion.** We train the 1B model using the *tuned* batch size $B_0 = 256$, which was obtained on a smaller 124M model. We set the sequence length $S_0 = 1024$ with Frank–Wolfe stepsize $\beta_0 = 3.6 \cdot 10^{-4}$ over the entire horizon $T_1 = 10.8\text{B}$ (in light blue). This configuration is motivated by hyperparameter transfer results under the $\mu$P framework, where the hyperparameters tuned for a smaller model are used when training a larger model.

- **Fixed *large*-batch Scion.** We train the 1B model using a larger batch size–sequence-length product. In particular, we consider two settings. For $B_1 = 512, S_1 = 2048$, we evaluate two baselines trained from the beginning over the full token budget $T_1 = 10.8\text{B}$: one with Frank–Wolfe stepsize $\beta_0$ (in orange), suggested by the $\mu$P framework, and one with stepsize $\beta_1$ (in pink), suggested by (10). For $B_2 = 1024$ and $S_2 = 1024$, we again train from the beginning

---

[8]The batch size, sequence length, and Frank–Wolfe stepsize used in the first and second training stages are determined using (9) and (10) and reported in Table 3.

*Table C.1.* Estimated $L$ constant for various model configurations.

| n_embd \ n_layer | 3 | 6 | 9 | 12 | 15 | 18 | 21 | 24 | 27 | 30 |
|---|---|---|---|---|---|---|---|---|---|---|
| **384** | – | 5.3 | 6.2 | 6.5 | 6.2 | 18 | 7.6 | – | 27 | 7.84 |
| **576** | – | 6.4 | 7.7 | 7.0 | 7.4 | – | 7.5 | 8.5 | 7.9 | 8.2 |
| **768** | 6.1 | 6.5 | 6.9 | 7.6 | 8.3 | 18 | 9.9 | 8.5 | 8.7 | 10.0 |
| **1152** | – | – | – | 8.8 | 9.4 | 10.8 | 9.5 | – | – | – |
| **1536** | 3 | 7.9 | 9.2 | 12 | 9.9 | – | – | – | – | – |
| **2304** | – | 9.9 | – | – | – | – | – | – | – | – |

over the full token budget $T_1 = 10.8\text{B}$, and consider two baselines: one with stepsize $\beta_0$ (in blue), suggested by the $\mu$P framework, and one with stepsize $\beta_1$ (in green), suggested by (10).

## C. Empirical Verification of Assumptions

### C.1. Verification of Assumption 4

In this section, we provide the evolution of the empirical variance throughout the training when varying the batch size and sequence length when training a base 124M model. When we vary the batch size, we keep the sequence length equal 1024; when we vary the sequence length, we keep the batch size equal 512. To approximate the full gradient, we sample a mini-batch gradient of size 32768. In Figure B.1, we demonstrate that after a short initial phase (up to 1k iterations) the empirical variance stabilizes and fluctuates around the average, suggesting that the variance is fixed during most of the training.

### C.2. Verification of Assumptions 1-3 when Varying Model Configuration

In this section, we provide the estimations of problem-dependent constants $L, \mu, \rho$, varying `n_embd` and `n_layer`, while keeping `n_head`=6. We measure the constants for several configurations to cover a broader range of configurations. Due to extensive requirements on memory and time resources, we do not provide the measurements for all possible configurations. Note that the norms are defined as follows

$$\|x\| = \max_{\ell \in [N]} \|x_\ell\|_\ell, \quad \|x\|_* = \sum_{\ell=1}^{N} \|x_\ell\|_{*,\ell}, \tag{14}$$

where we use infinity norm for the embedding/head layer, and spectral norm for all hidden layers.

C.2.1. ESTIMATING THE SMOOTHNESS CONSTANT $L$

First, we measure the smoothness constant $L$ using the following estimation

$$\frac{\|g(x_k;\xi_k) - g(x_{k-1};\xi_{k-1})\|_*}{\|x_k - x_{k-1}\|},$$

where $g(x_k;\xi_k), g(x_{k-1};\xi_{k-1})$ are mini-batch gradients at two consecutive iterations, while the norms are defined in (14). Based on the results in Table C.1, we fit a power law with shifts of the form

$$L(\texttt{n\_layer}, \texttt{n\_embd}) = C(\texttt{n\_layer} + a_0)^\nu (\texttt{n\_embd} + b_0)^\gamma.$$

We fit the parameters of the power law in log-space, using least squares with `soft_l1` loss from `scipy.optimize` (Virtanen et al., 2020). The fit provides the following approximations for the constants of the power law:

$$C = 0.4, \quad \nu = 0.2, \quad \gamma = 0.35, \quad a_0 = 0.7, \quad b_0 = 126.$$

C.2.2. ESTIMATING THE NORM EQUIVALENCE CONSTANT $\rho$

Now we measure the norm equivalence constant $\rho$. We observed that the $\rho$ constant changes not only with the model size but also with the batch size and sequence length. To measure it, we run Scion with batch size $512$ and sequence length

*Table C.2.* Estimated $\rho$ constant for various model configurations.

| n_embd \ n_layer | 3 | 6 | 9 | 12 | 15 | 18 | 21 | 24 | 27 | 30 |
|---|---|---|---|---|---|---|---|---|---|---|
| 384 | – | 35.5 | 41.3 | 48.2 | 48.7 | – | 50.7 | – | – | 58.0 |
| 576 | – | 42.1 | 53.8 | 61.1 | 62.9 | – | 64.2 | 64.6 | 66.2 | 68.6 |
| 768 | 31.2 | 52.6 | 64.1 | 67.1 | 72.4 | – | 80.2 | 87.0 | 86.3 | 89.2 |
| 1152 | – | – | – | 76.6 | 81.1 | 87.3 | — | – | – | – |
| 1536 | – | 67.5 | 77.4 | – | – | – | – | – | – | – |
| 2304 | – | – | – | – | – | – | – | – | – | – |

*Table C.3.* Estimated $\rho$ constant for a configuration with 6 layers and 768 embedding dimension when varying the batch size.

| batch_size | 256 | 512 | 1024 | 2048 | 4096 |
|---|---|---|---|---|---|
| $\rho$ | 48.9 | 52.6 | 55.0 | 56.4 | 57.9 |

1024, and the Frank–Wolfe stepsize $\beta = 3.6 \cdot 10^{-4}$. We estimate the $\rho$ constant as follows

$$\frac{\|g(x_k; \xi_k) - G(x_k; \Xi_k)\|_*}{\|g(x_k; \xi_k) - G(x_k; \Xi_k)\|_2},$$

where $g(x_k; \xi_k)$ is a mini-batch gradient of size 512, while $G(x_k; \Xi_k)$ is a mini-batch gradient of size 8192, which serves as an approximation of the full gradient.

We also observe that the constant $\rho$ significantly changes with the batch size. Therefore, we measured how $\rho$ changes with batch size for a configuration with 6 layers and 768 embedding dimension in Table C.3.

Based on the results in Table C.2 and Table C.3, we fit the parameters of the power law of the form

$$\rho(\texttt{n\_layer}, \texttt{n\_embd}, \texttt{batch\_size}) = C(\texttt{n\_layer} + a_0)^\nu (\texttt{n\_embd} + b_0)^\gamma (\texttt{batch\_size} + c_0)^\delta$$

in log-space, using least squares with `soft_l1` loss from `scipy.optimize`. The fit provides the following approximations for the constants of the power law:

$$C = 4.1, \quad a_0 = -2.7, \quad \nu = 0.25, \quad b_0 = -250.8, \quad \gamma = 0.3, \quad c_0 = -9.4, \quad \delta = 0.1.$$

### C.2.3. ESTIMATING KURDYKA–ŁOJASIEWICZ CONSTANT $\mu$

Finally, we measure the KL constant $\mu$ by tracking the dual gradient norm and train loss (norms are defined in (14)). Then, we fit a linear regression with Huber loss, robust to outliers. The slope of the linear fit serves as an approximation of $\mu$ constant. Based on the results in Table C.4, we fit the parameters of the power law of the form

$$\mu(\texttt{n\_layer}, \texttt{n\_embd}) = C(\texttt{n\_layer} + a_0)^\nu (\texttt{n\_embd} + b_0)^\gamma.$$

in log-space, using least squares with `soft_l1` loss from `scipy.optimize`. The fit provides the following approximations for the constants of the power law:

$$C = 5.2, \quad \nu = 0.2, \quad \gamma = 0, \quad a_0 = 1.7, \quad b_0 = -384.$$

## D. Additional Experiments

### D.1. Momentum Parameter Transfer

From Table 2, we know that for a base 124M model, the optimal set of hyperparameters is $B = 256, S = 1024, \beta = 3.6 \cdot 10^{-4}$ under token budget $T = 1.3B$ (TPP 10.8). We want to use these parameters in the BST rule to obtain them for a larger training horizon or model size. In this section, the reported train losses are averaged over 3 random seeds (only for 124M) and smoothed using a running average with a window size of 500 (for both 124M and 1B models). We ignore the requirement that $B$ be a power of two in this set of experiments.

*Table C.4.* Estimated Kurdyka–Łojasiewicz constant $\mu$ for various model configurations.

| n_embd \ n_layer | 3 | 6 | 9 | 12 | 15 | 18 | 21 | 24 | 27 | 30 |
|---|---|---|---|---|---|---|---|---|---|---|
| **384** | – | 3.4 | 3.1 | 3.1 | 3.0 | – | 2.9 | – | – | 2.7 |
| **576** | – | 3.3 | 3.2 | 3.0 | 2.9 | – | 2.7 | 2.6 | 2.5 | 2.4 |
| **768** | 3.7 | 3.2 | 3.0 | 2.9 | 2.8 | – | 2.6 | 2.5 | 2.3 | 2.4 |
| **1152** | – | – | – | 2.7 | 2.7 | 2.8 | 2.6 | – | 2.5 | – |
| **1536** | – | 3.2 | 2.9 | – | 2.9 | 3.0 | – | – | – | – |
| **2304** | – | 3.6 | – | – | – | – | – | – | – | – |

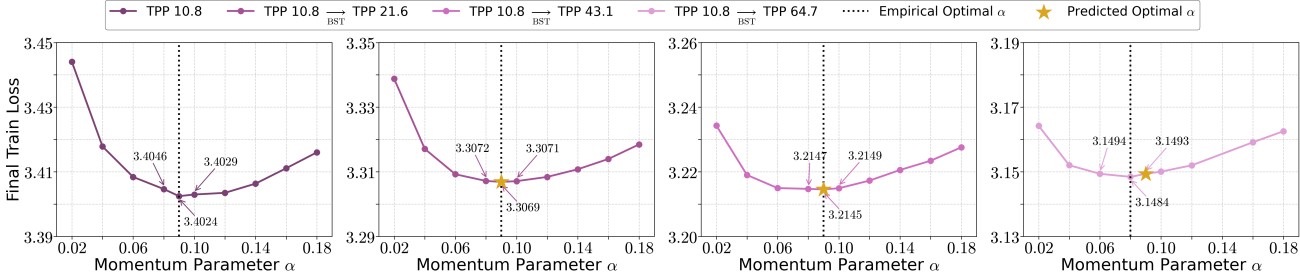

*Figure D.1.* Smoothed train-loss trajectories for the 124M model when varying the Frank–Wolfe stepsize $\beta$ under different token budgets: 2.7B tokens (**left**), 5.3B tokens (**center**), and 8.0B tokens (**right**). We average the train loss over 3 random seeds and report the moving average in a window of size 500. We observe that the momentum parameter $\alpha$ transfers under BST scaling.

### D.1.1. 124M Model under Larger Token Budget

In this section, we report the pretraining results for 124M model under increased token budgets $(i)$ $T = 2.7$B (TPP 21.6), $(ii)$ $T = 5.3$B (TPP 43.1), and $(iii)$ $T = 8.0$B (TPP 64.7). We set a batch size for longer horizons using (9): $B = 416$ for $T = 2.7$B, $B = 672$ for $T = 5.3$B, and $B = 896$ for $T = 8.0$B, using estimates of problem-dependent constants from (3). Momentum and sequence length are set to $\alpha = 0.1$ and $S = 1024$, respectively.

We test the BST rule for predicting the optimal value of the momentum parameter $\alpha$. According to the BST rule, $\alpha$ should transfer. Empirical results in Figure D.1 support this claim. For all values of the token budget, the optimal $\alpha$ is close to 0.09.

### D.1.2. Increasing Model Size

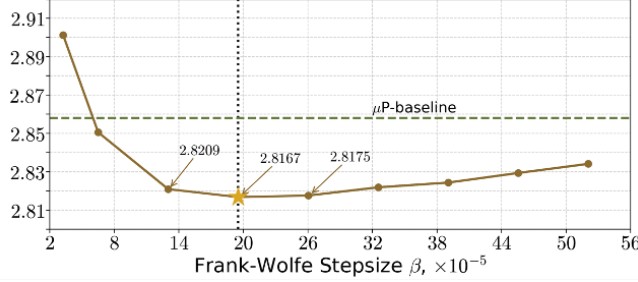

*Figure D.2.* Final train loss for the 1B model when varying the momentum parameter $\alpha$ under a token budget of 10.8B tokens (TPP 10.8). We report the final train loss, smoothed in the window of size 500. We observe that the BST scaling rule predicts a good estimate for optimal $\alpha$ when transferring from a smaller 124M model to a larger 1B model.

Now we want to train a 1B model with batch size $B = 1120, S = 1024$ under token budget 10.8B (TPP 10.8). In this setup, we test the predictive power of the BST scaling rule when we change the model size. The value of the batch size is set according to (9), using estimates from Table 3. We expect the optimal Frank–Wolfe stepsize to be close to $1.95 \cdot 10^{-4}$, while the momentum parameter to be close to 0.09. We report the results in Figure D.2. We observe that the BST rule provides a good estimation for both the optimal momentum $\alpha$ and Frank–Wolfe stepsize $\beta$, when increasing the model

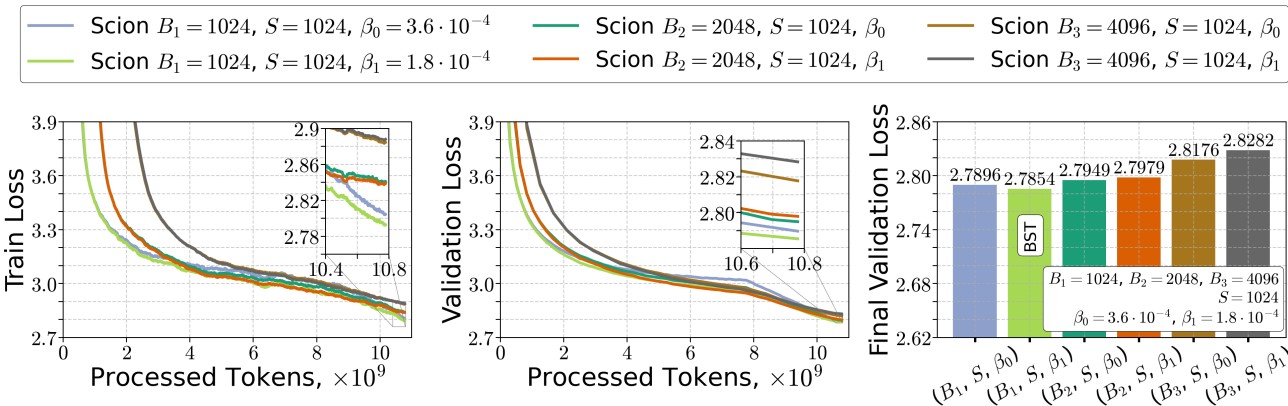

*Figure D.3.* Comparison of fixed large batch size strategies when training a 1B model. The validation loss is evaluated with a smaller sequence length 1024. Scion with a batch size of 1024 suggested by our BST scaling rule achieves the best performance compared to other baselines with batch sizes 2048 and 4096. The values of batch sizes $B_{1,2,3}$, sequence lengths $S$, and Frank–Wolfe stepsizes $\beta_{0,1}$ are given in the legends. The notation $(B_{1,2,3}, S, \beta_{0,1})$ characterizes which batch size, sequence length, and Frank–Wolfe stepsize are used for the particular setup, respectively. The notation BST indicates the rule used to select the $B$, $S$, and $\beta$.

size.

### D.2. Increasing Batch Size Further Does Not Help

Next, we investigate whether increasing the product $BS$ by 8 or 16 times (when fixing a train sequence length 1024, this results in batch sizes 2048 and 4096) yields additional benefits when training a larger 1B model. All experiments are conducted with fixed training and validation sequence lengths of 1024. We report results using both Frank–Wolfe stepsizes suggested by the $\mu$P framework and those determined by our BST scaling rule.

The results are shown in Figure D.3. We observe that Scion with a batch size of 1024 outperforms the baselines with batch sizes 2048 and 4096. This finding suggests that our BST scaling rule, which prescribes how to scale the product $BS$, provides a reliable practical guideline. Increasing the product $BS$ beyond this recommendation does not yield further performance gains. In particular, the validation loss for Scion with batch size 2048 worsens by approximately 0.005-0.01, while for batch size 4096 the degradation is more pronounced, around 0.03-0.04.

### D.3. Additional Baselines in Experiments from Section 6.4

In this section, we add additional baselines to the setting from Section 6.4. The idea behind the two new baselines is the following. The literature on the learning theory suggests that the excess risk should decay as $\sim \frac{1}{\sqrt{T}}$ under standard convexity (Shalev-Shwartz et al., 2009; Liu & Tong, 2024), which is the closest setting to $\mu$-KL case due to the relation between $\mu$-KL condition and $\zeta$-quasar convexity, described after Assumption 3. This hypothesizes that we need to keep the optimization error close to the excess risk. In particular, if we find that for a small model the dominating term in (3) is the first one, then we control it as follows

$$\frac{LB_0 S_0}{\mu^2 T_0} \sim \frac{\varepsilon_0}{\sqrt{T_0}}.$$

We want to choose parameters $B_1$ and $S_1$ such that the same approximation holds for a larger model. This gives another recipe on how to increase the batch size and sequence length:

$$\frac{B_0 S_0 / T_0}{B_1 S_1 / T_1} = \frac{1/\sqrt{T_0}}{1/\sqrt{T_1}} \Rightarrow B_1 S_1 = B_0 S_0 \frac{\sqrt{T_1}}{\sqrt{T_0}}. \tag{15}$$

From (10), we obtain that we should choose the Frank–Wolfe stepsize of the form

$$\beta_1 = \beta_0 \frac{B_1 S_1}{B_0 S_0} \frac{T_0}{T_1} = \frac{\sqrt{T_0}}{\sqrt{T_1}}. \tag{16}$$

For a 1B model, this means that we should increase the product $BS$ by a factor $\sqrt{8}$, while decreasing the Frank–Wolfe stepsize by a factor $1/\sqrt{8}$. Performing all derivations, this gives the values of batch size 736, sequence length 1024, and the

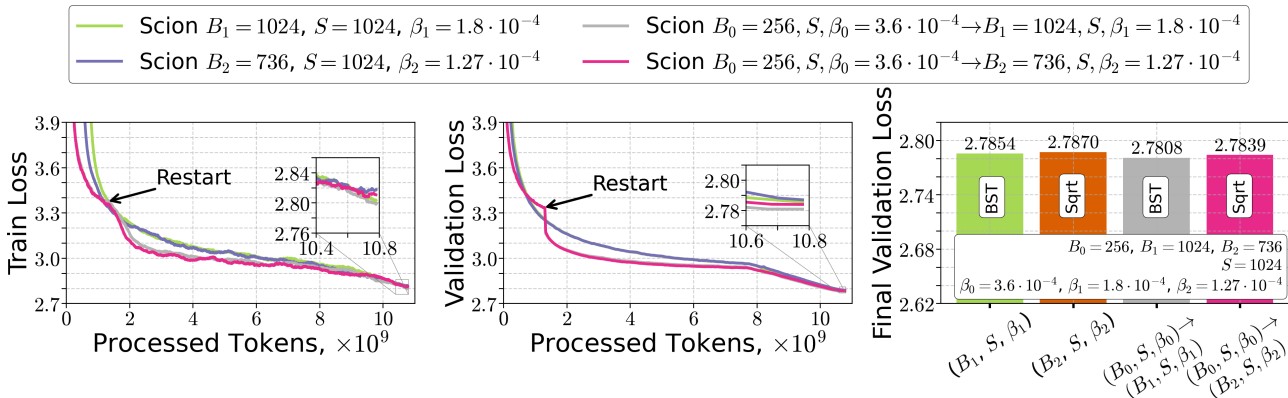

*Figure D.4.* Comparison of two strategies: BST scaling rule, where $BS \sim T^{2/3}$ (fixed 1024 batch size with $\beta_1 = 1.8 \cdot 10^{-4}$ in light green and restarted version in gray) and square-root rule (15), where $BS \sim T^{1/2}$ (fixed 736 batch size with $\beta_2 = 1.27 \cdot 10^{-4}$ in orange and restarted version in pink). The validation and training sequence lengths are fixed to 1024. Scion with batch size 1024, used either from the beginning or after a restart, achieves slightly better performance than the square-root-rule baselines in (15). The notation $(B_{1,2}, S, \beta_{0,1})$ characterizes which batch size, sequence length, and Frank–Wolfe stepsize are used for the particular setup, respectively. The notation $(B_0, S, \beta_0) \to (B_{1,2}, S, \beta_{1,2})$ characterizes how parameters of Scion change after restart (e.g., batch size increases from $B_0$ to $B_{1,2}$), respectively.

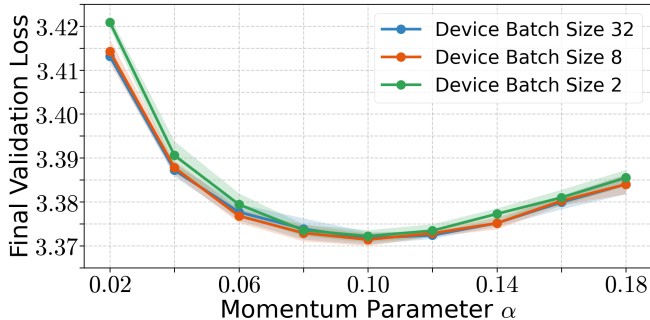

*Figure D.5.* Final performance of 124M model with $B = 256, S = 1024, \beta = 3.6 \cdot 10^{-4}$ and varying the momentum parameter $\alpha$ and device batch size.

Frank–Wolfe stepsize $1.27 \cdot 10^{-4}$. In Figure D.4, we add two more baselines: one when we do a restart with the parameters $736, 1024, 1.27 \cdot 10^{-4}$ after a 1.3B token budget, and another one where we use the parameters $736, 1024, 1.27 \cdot 10^{-4}$ from the beginning. These ideas are closely aligned with prior work Compagnoni et al. (2025b;a); Mlodozeniec et al. (2026), which also propose to rescale the weight decay (equivalent of our Frank–Wolfe stepsize) following the square-root rule.

We observe that the new baselines are also competitive in practice. However, more aggressive BST scaling rules in (9) and (10) provide slightly better results: the restarted version of BST baseline achieves the best validation loss, while a fixed batch BST baseline slightly outperforms square-root fixed batch baseline. This further supports that our theory-inspired BST scaling rule is predictive and can be used in practice. The notation Sqrt or BST indicates the rule used to select the Frank–Wolfe stepsize.

### D.4. Effect of the Device Batch Size

In this section, we evaluate how the device batch size affects the final performance of the 124M model. Note that using a smaller device batch size within a fixed global batch size results in more gradient accumulation steps. In Figure D.5, we report the model's final validation loss as we vary the device batch size and the momentum parameter. We observe that for device batch sizes 8 and 32, the performance is closely aligned. The quantization errors become slightly visible for a device batch size of 2 when using extreme values of momentum far from the optimal value. However, around the optimal momentum parameter, the difference is within one standard deviation.

# E. In-Expectation Convergence Proofs for SCG

The proof structure is inspired by the analysis of the first-order stochastic trust-region method with momentum under star-convexity by Kovalev (2025).

**Lemma 1.** Let assumptions (A1) and (A3) hold. Assume that $x_0$ and $\eta$ are chosen such that

$$2\|x_0\| \leq \eta, \quad \beta = \frac{c}{K}, \quad \text{and} \quad K \geq 2c. \tag{17}$$

Let $\{x_k\}$ be the iterates of Algorithm 1 . Then, the following inequalities hold for all $k \in \{0, 1 \ldots, K-1\}$

$$\eta - \|x_k\| \geq (1-\beta)^k \frac{\eta}{2}, \quad \|x_{k+1} - x_k\| \leq 2\beta\eta. \tag{18}$$

*Proof.* We show by induction $k$ that

$$\|x_k\| \leq (1-\beta)^k \frac{\eta}{2} + \eta(1 - (1-\beta)^k)$$

holds. The base of induction is $k = 0$. Note that $\|x_0\| \leq \frac{\eta}{2}$ holds by the choice of $\eta$ and $x_0$ in (17). Assume that inequalities hold for some $k \in \{0, 1, \ldots K-2\}$. We show that they also hold at iteration $k + 1$. Indeed, we have

$$\|x_{k+1}\| \overset{(a)}{=} \|(1-\beta)x_k + \beta\eta d_{k+1}\| \overset{(b)}{\leq} (1-\beta)\|x_k\| + \beta\eta\|d_{k+1}\|$$

$$\overset{(c)}{\leq} (1-\beta)\left((1-\beta)^k \frac{\eta}{2} + \eta(1-(1-\beta)^k)\right) + \beta\eta$$

$$= (1-\beta)^{k+1}\frac{\eta}{2} + \eta((1-\beta) - (1-\beta)^{k+1} + \beta) = (1-\beta)^{k+1}\frac{\eta}{2} + \eta(1-(1-\beta)^{k+1}),$$

where (a) uses the update step; (b) uses the triangle inequality; (c) uses the restriction on $d_{k+1}$ in and induction hypothesis. This concludes the induction step and proves the first inequality in (18) for all $k \in \{0, 1, \ldots, K\}$. We can lower bound $(1-\beta)^K$ using the inequality $\log(1-y) \geq -y - y^2$ for all $y \in [0, 0.5]$ and $K \geq 2c$ as follows

$$\log((1-\beta)^K) = K\log(1-\beta) \geq K(-\beta - \beta^2) = -c - \frac{c^2}{K} \geq -\frac{3c}{2}. \tag{19}$$

This implies that $(1-\beta)^K \geq e^{-3c/2}$. Using the obtained bound, we derive for all $k \in \{0, 1, \ldots, K\}$

$$\|x_k\| \leq (1-\beta)^K \frac{\eta}{2} + \eta(1 - (1-\beta)^K) \leq \eta - \frac{\eta}{2}e^{-3c/2}. \tag{20}$$

Now we prove the last inequality in (18). We have

$$\|x_{k+1} - x_k\| \overset{(a)}{=} \| - \beta x_k + \beta\eta d_{k+1}\|$$

$$\overset{(b)}{\leq} \beta\|x_k\| + \beta\eta\|d_{k+1}\|$$

$$\overset{(c)}{\leq} \beta((1-\beta)^K \frac{\eta}{2} + \eta(1-(1-\beta)^K)) + \beta\eta$$

$$= \beta\eta((1-\beta)^K/2 + 1 - (1-\beta)^K + 1)$$

$$= \beta\eta(2 - (1-\beta)^K/2) \leq 2\beta\eta,$$

where (a) uses the update rule; (b) uses the triangle inequality; (c) uses the previous inequality and the restriction on $d_{k+1}$. □

**Lemma 2.** Let Assumptions (A1), (A2) and (A4) hold. Let $m_0 = g(x_0; \xi_0)$, then the iterates of Algorithm 1 satisfy the following inequality:

$$\mathbb{E}[\|m_{k+1} - \nabla f(x_k)\|_*] \leq (1-\alpha)^k \rho\sigma + \frac{2L\beta\eta}{\alpha} + \rho\sigma\sqrt{\alpha}.$$

*Proof.* We can express $m_{k+1} - \nabla f(x_k)$ as follows using the definition of the momentum buffer in Algorithm 1

$$
\begin{aligned}
m_{k+1} - \nabla f(x_k) &= (1-\alpha)m_k + \alpha g(x_k; \xi_k) - \nabla f(x_k) \\
&= (1-\alpha)(m_k - \nabla f(x_{k-1})) + \alpha(g(x_k; \xi_k) - \nabla f(x_k)) \\
&\quad + (1-\alpha)(\nabla f(x_{k-1}) - \nabla f(x_k)).
\end{aligned}
$$

This implies the following for all $k \geq 0$:

$$
m_{k+1} - \nabla f(x_k) = (1-\alpha)^k(m_1 - \nabla f(x_0)) + \sum_{i=0}^{k-1}(1-\alpha)^{k-i}(\nabla f(x_i) - \nabla f(x_{i+1}))
$$

$$
+ \sum_{i=1}^{k}\alpha(1-\alpha)^{k-i}(g(x_i, \xi_i) - \nabla f(x_i)).
$$

Using this decomposition, we can upper-bound $\|m_{k+1} - \nabla f(x_k)\|_*$ as follows

$$
\begin{aligned}
\|m_{k+1} - \nabla f(x_k)\|_* &\overset{(a)}{\leq} (1-\alpha)^k\|m_1 - \nabla f(x_0)\|_* + \sum_{i=0}^{k-1}(1-\alpha)^{k-i}\|\nabla f(x_i) - \nabla f(x_{i+1})\|_* \\
&\quad + \|\sum_{i=1}^{k}\alpha(1-\alpha)^{k-i}(g(x_i, \xi_i) - \nabla f(x_i))\|_* \\
&\overset{(b)}{\leq} (1-\alpha)^k\|m_1 - \nabla f(x_0)\|_* \\
&\quad + \sum_{i=0}^{k-1}(1-\alpha)^{k-i}2L\beta\eta + \|\sum_{i=1}^{k}\alpha(1-\alpha)^{k-i}(g(x_i, \xi_i) - \nabla f(x_i))\|_* \\
&\overset{(c)}{\leq} (1-\alpha)^k\rho\|m_1 - \nabla f(x_0)\|_2 + \sum_{i=0}^{k-1}(1-\alpha)^{k-i}2L\beta\eta \\
&\quad + \rho\|\sum_{i=1}^{k}\alpha(1-\alpha)^{k-i}(g(x_i, \xi_i) - \nabla f(x_i))\|_2,
\end{aligned}
$$

where (a) uses the triangle inequality; (b) uses (A1) and Lemma 1 with $L, \beta, \eta$; (c) uses (A2). Next, we take the full expectation and get

$$
\begin{aligned}
\mathbb{E}\left[\|m_{k+1} - \nabla f(x_k)\|_*\right] &\leq (1-\alpha)^k\rho\mathbb{E}[\|m_1 - \nabla f(x_0)\|_2] + \sum_{i=0}^{k-1}(1-\alpha)^{k-i}2L\beta\eta \\
&\quad + \rho\mathbb{E}\left[\|\sum_{i=1}^{k}\alpha(1-\alpha)^{k-i}(g(x_i, \xi_i) - \nabla f(x_i))\|_2\right] \\
&\overset{(a)}{\leq} (1-\alpha)^k\rho\sqrt{\mathbb{E}[\|m_1 - \nabla f(x_0)\|_2^2]} + \sum_{i=0}^{k-1}(1-\alpha)^{k-i}2L\beta\eta \\
&\quad + \rho\sqrt{\mathbb{E}\left[\|\sum_{i=1}^{k}\alpha(1-\alpha)^{k-i}(g(x_i, \xi_i) - \nabla f(x_i))\|_2^2\right]} \\
&\overset{(b)}{\leq} (1-\alpha)^k\rho\sigma + \sum_{i=0}^{k-1}(1-\alpha)^{k-i}2L\beta\eta + \alpha\rho\sigma\sqrt{\sum_{i=1}^{k}(1-\alpha)^{2(k-i)}} \\
&\leq (1-\alpha)^k\rho\sigma + \frac{2L\beta\eta}{\alpha} + \sqrt{\alpha}\rho\sigma,
\end{aligned}
$$

where (a) uses Jensen's inequality; (b) uses (A4) and the fact that samples $\xi_i \sim \mathcal{D}$ are independent. $\square$

**Theorem 2** (Full statement of Theorem 1). *Let Assumption* (A1), (A2), (A3), (A4) *hold. Let* $m_0 = g(x_0; \xi_0)$ *and* $c > 0$. *Let the parameters of Algorithm 1 are chosen as follows*

$$\beta = \frac{c}{K}, \quad \eta = \frac{2e^{3c/2}}{\mu c} \log\left(\frac{2(f(x_0) - f^\star)}{\varepsilon}\right), \quad 2\|x_0\| \leq \eta, \tag{21}$$

*and*

$$\alpha = \min\left\{1, \frac{(\varepsilon\mu)^2}{(32\rho\sigma)^2 e^{3c}}\right\}, \tag{22}$$

$$K = \max\left[2c, \max\left\{\frac{1}{2}, \frac{128Le^{3c}}{\varepsilon\mu^2}, \frac{32\rho\sigma e^{3c/2}}{\varepsilon\mu}, \frac{128Le^{6c}(32\rho\sigma)^2}{\mu(\varepsilon\mu)^3}, \frac{(32\rho\sigma e^{3c/2})^3}{(\varepsilon\mu)^3}\right\} \log\left(\frac{2(f(x_0) - f^\star)}{\varepsilon}\right)\right].$$

*Then the output of Algorithm 1 after $K$ iterations satisfies $\mathbb{E}[f(x_K) - f^\star] \leq \varepsilon$.*

**Remark 3.** The choice of $\eta \sim \log\left(\frac{2(f(x_0)-f^\star)}{\varepsilon}\right)$ and $2\|x_0\| \leq \eta$ ensures a sufficient contraction factor in front of $f(x_k) - f^\star$ in the proof. We also note that all iterates produced by Algorithm 1 have a bounded norm by $\eta$. However, we do not make any explicit assumptions about $\arg\min_{x \in \mathcal{X}} f(x)$, e.g., we do not assume its existence or boundedness of its norm by $\eta$. Therefore, for a fixed choice of $\varepsilon$ it is possible that an optimizer has norm larger than $\eta$ while $\mathbb{E}[f(x_K) - f^\star] \leq \varepsilon$.

*Proof.* Let $u_k = \arg\min_{u \in \mathcal{X}} \langle \nabla f(x_k), u \rangle$ s.t. $\|u\| \leq 1$. Then we have

$$
\begin{aligned}
f(x_{k+1}) &\overset{(a)}{\leq} f(x_k) + \langle \nabla f(x_k), x_{k+1} - x_k \rangle + \frac{1}{2}L\|x_{k+1} - x_k\|^2 \\
&\overset{(b)}{=} f(x_k) + \langle \nabla f(x_k), -\beta x_k + \beta\eta d_{k+1}\rangle + 2L\beta^2\eta^2 \\
&= f(x_k) - \beta\langle\nabla f(x_k), x_k\rangle + \beta\eta\langle\nabla f(x_k) - m_{k+1}, d_{k+1}\rangle + \beta\eta\langle m_{k+1}, d_{k+1}\rangle + 2L\beta^2\eta^2 \\
&\overset{(c)}{\leq} f(x_k) - \beta\langle\nabla f(x_k), x_k\rangle + \beta\eta\langle\nabla f(x_k) - m_{k+1}, d_{k+1}\rangle + \beta\eta\langle m_{k+1}, u_k\rangle + 2L\beta^2\eta^2 \\
&\overset{(d)}{=} f(x_k) - \beta\langle\nabla f(x_k), x_k\rangle + \beta\eta\langle\nabla f(x_k) - m_{k+1}, d_{k+1} - u_k\rangle - \beta\eta\|\nabla f(x_k)\|_* + 2L\beta^2\eta^2 \\
&\overset{(e)}{\leq} f(x_k) + \beta\|\nabla f(x_k)\|_* \cdot \|x_k\| + 2\beta\eta\|\nabla f(x_k) - m_{k+1}\|_* - \beta\eta\|\nabla f(x_k)\|_* + 2L\beta^2\eta^2 \\
&= f(x_k) - \beta\|\nabla f(x_k)\|_*(\eta - \|x_k\|) + 2\beta\eta\|m_{k+1} - \nabla f(x_k)\|_* + 2L\beta^2\eta^2 \\
&\overset{(f)}{\leq} f(x_k) - \frac{\beta\eta\mu}{2}e^{-3c/2}(f(x_k) - f^\star) + 2\beta\eta\|m_{k+1} - \nabla f(x_k)\|_* + 2L\beta^2\eta^2, \tag{23}
\end{aligned}
$$

where (a) uses (A1); (b) uses the update step and Lemma 1; (c) uses the optimality of $d_{k+1}$; (d) uses $\langle\nabla f(x_k), u_k\rangle = -\|\nabla f(x_k)\|_*$; (e) uses Cauchy-Schwarz and $\|d_{k+1}\|, \|u_k\| \leq 1$; (f) uses Lemma 1, (A3), and (20). With the assumption that $m_0 = g(x_0; \xi_0)$, we have from Lemma 2 that

$$\mathbb{E}[\|m_{k+1} - \nabla f(x_k)\|_*] \leq (1 - \alpha)^k \rho\sigma + \frac{2L\beta\eta}{\alpha} + \rho\sigma\sqrt{\alpha}.$$

Taking the expectation from (23) and using this bound and Lemma 1, we derive

$$\mathbb{E}[f(x_{k+1}) - f^\star] \leq \left(1 - \frac{\mu\beta\eta}{2e^{3c/2}}\right)\mathbb{E}[f(x_k) - f^\star] + (1-\alpha)^k 2\beta\eta\rho\sigma + \frac{4L\beta^2\eta^2}{\alpha} + 2\beta\eta\rho\sigma\sqrt{\alpha}$$
$$+ 2L\beta^2\eta^2. \tag{24}$$

The contraction factor $1 - \frac{\mu\beta\eta}{2e^{3c/2}} \in (0, 1)$ by the choice of $K \geq \frac{1}{2}\log\left(\frac{2(f(x_0)-f^*)}{\varepsilon}\right)$. Unrolling this recursion for all iterations $k \in \{0, 1, \ldots, K-1\}$ and using the bound for the geometric series, we guarantee progress such that

$$\mathbb{E}[f(x_K) - f(x^\star)] \leq \left(1 - \frac{\mu\beta\eta}{2e^{3c/2}}\right)^K (f(x_0) - f(x^\star)) + \frac{2\beta\eta\rho\sigma}{\alpha} + \frac{4\rho\sigma\sqrt{\alpha}}{\mu}e^{3c/2}$$
$$+ \frac{4L\beta\eta}{\mu}e^{3c/2} + \frac{8L\beta\eta}{\alpha\mu}e^{3c/2}. \tag{25}$$

Now we need to bound each of the terms proportionally to $\varepsilon$ using the choice of parameters $\eta, \alpha, \beta, K$ from (21) and (22). First, we want

$$4\rho\sigma\frac{\sqrt{\alpha}}{\mu}e^{3c/2} \leq \frac{\varepsilon}{8} \Rightarrow \alpha \leq \frac{(\varepsilon\mu)^2}{(32\rho\sigma)^2 e^{3c}}.$$

We can satisfy the above bound with the choice of $\alpha$ such that

$$\alpha = \min\left\{1, \frac{(\varepsilon\mu)^2}{(32\rho\sigma)^2 e^{3c}}\right\}, \tag{26}$$

which is exactly the choice of $\alpha$ in (22). Next, we want

$$\frac{8Le^{3c/2}}{\mu}\frac{\beta\eta}{\alpha} \leq \frac{\varepsilon}{8} \Rightarrow \beta = \frac{c}{K} \leq \frac{\varepsilon\mu\alpha}{64Ln e^{3c/2}} \overset{(a)}{\leq} \min\left\{\frac{\varepsilon\mu}{64Ln e^{3c/2}}, \frac{(\varepsilon\mu)^3}{64Ln e^{9c/2}(32\rho\sigma)^2}\right\},$$

where (a) uses (26). The above can be satisfied if we choose $K$ such that

$$K \geq \max\left\{\frac{64Ln c e^{3c/2}}{\varepsilon\mu}, \frac{64Ln c e^{9c/2}(32\rho\sigma)^2}{(\varepsilon\mu)^3}\right\},$$
$$\overset{(a)}{=} \max\left\{\frac{128Le^{3c}}{\varepsilon\mu^2}, \frac{128Le^{6c}(32\rho\sigma)^2}{\mu(\varepsilon\mu)^3}\right\} \cdot \log\left(\frac{2(f(x_0) - f^\star)}{\varepsilon}\right), \tag{27}$$

where (a) uses the value of $\eta$. This choice of $K$ is satisfied by the choice in (22). Moving on, we want

$$\frac{4L\beta\eta e^{3c/2}}{\mu} \leq \frac{\varepsilon}{8} \Rightarrow \beta = \frac{c}{K} \leq \frac{\varepsilon\mu}{32Ln e^{3c/2}}. \tag{28}$$

We can satisfy the right inequality above if we choose $K$ such that

$$K \geq \frac{32Ln c e^{3c/2}}{\varepsilon\mu} \overset{(a)}{=} \frac{64Le^{3c}}{\mu^2\varepsilon}\log\left(\frac{2(f(x_0) - f^\star)}{\varepsilon}\right), \tag{29}$$

where (a) uses the value of $\eta$ in (21). This choice of $K$ is satisfied by the choice in (22). Finally, we want

$$2\rho\sigma\frac{\beta\eta}{\alpha} \leq \frac{\varepsilon}{8} \Rightarrow \beta = \frac{c}{K} \leq \frac{\varepsilon\alpha}{16\rho\sigma\eta} \overset{(a)}{\leq} \min\left\{\frac{\varepsilon}{16\rho\sigma\eta}, \frac{\varepsilon(\varepsilon\mu)^2}{16\rho\sigma\eta(32\rho\sigma)^2 e^{3c}}\right\},$$

where (a) uses (26). The above inequality is satisfied with the choice of $K$ such that

$$K \geq \max\left\{\frac{16\rho\sigma\eta c}{\varepsilon}, \frac{16\rho\sigma\eta c(32\rho\sigma)^2 e^{3c}}{\varepsilon(\varepsilon\mu)^2}\right\} \tag{30}$$
$$\overset{(a)}{=} \max\left\{\frac{32\rho\sigma e^{3c/2}}{\varepsilon\mu}, \frac{(32\rho\sigma e^{3c/2})^3}{(\varepsilon\mu)^3}\right\}\log\left(\frac{2(f(x_0) - f^\star)}{\varepsilon}\right),$$

where (a) uses the value of $\eta$ in (21). This bound on $K$ is satisfied by the choice in (21). A combination of (27), (29), (30) gives the choice of $K$ in (22):

$$K = \max\left\{\frac{128Le^{3c}}{\varepsilon\mu^2}, \frac{32\rho\sigma e^{3c/2}}{\varepsilon\mu}, \frac{128Le^{6c}(32\rho\sigma)^2}{\mu(\varepsilon\mu)^3}, \frac{(32\rho\sigma e^{3c/2})^3}{(\varepsilon\mu)^3}\right\}\log\left(\frac{2(f(x_0) - f^\star)}{\varepsilon}\right). \tag{31}$$

Now we show that the choice of $K$, $\beta$, and $\eta$ ensures that the first term in (25) is smaller than $\varepsilon/2$. Let us show that

$$\left(1 - \frac{\mu\beta\eta}{2e^{3c/2}}\right)^K (f(x_0) - f^\star) \leq e^{-\mu\beta\eta e^{-3c/2}K/2}(f(x_0) - f^\star) \leq \frac{\varepsilon}{2}. \tag{32}$$

The last inequality is satisfied if the following condition holds:

$$\frac{\mu\beta\eta}{2e^{3c/2}}K \geq \log\left(\frac{2(f(x_0) - f^\star)}{\varepsilon}\right).$$

Plugging in the choice of $\beta = \frac{c}{K}$ and $\eta = \frac{2e^{3c/2}}{\mu c} \log\left(\frac{2(f(x_0) - f^\star)}{\varepsilon}\right)$, we obtain

$$\frac{\mu\beta\eta}{2e^{3c/2}} K = \frac{\mu}{2e^{3c/2}} \cdot \frac{c}{K} \cdot \frac{2e^{3c/2}}{\mu c} \log\left(\frac{2(f(x_0) - f^\star)}{\varepsilon}\right) \cdot K = \log\left(\frac{2(f(x_0) - f^\star)}{\varepsilon}\right).$$

Grouping the bounds together, we obtain that the choice of $K, \eta, \beta$ implies that

$$\mathbb{E}[f(x_K) - f^\star] \leq \frac{\varepsilon}{2} + 4 \cdot \frac{\varepsilon}{8} = \varepsilon.$$

$\square$

**Corollary 2** (Full statement of Corollary 1)**.** Under the setup of Theorem 2, let the token budget be large enough: $T \geq \max\left\{2cBS, \frac{BS}{2} \log\left(\frac{2(f(x_0) - f^\star)}{\varepsilon}\right)\right\}$. Then, running the algorithm with parameters from Theorem 2 for $K = T/BS$ iterations, we achieve the optimization error

$$\varepsilon = \max\left\{\frac{128LBSe^{3c}}{\mu^2 T}, \left(\frac{128Le^{6c}(32\rho\sigma_\star)^2}{\mu^4 T}\right)^{1/3}, \frac{32e^{3c/2}\rho\sigma_\star}{\mu(T^2 BS)^{1/6}}\right\}. \tag{33}$$

*Proof.* From Theorem 2, we have that to achieve the optimization error $\varepsilon$, we need to use $K$ iterations defined as

$$K = \max\left[\frac{128Le^{3c}}{\varepsilon\mu^2}, \frac{32e^{3c/2}\rho\sigma}{\varepsilon\mu}, \frac{128Le^{6c}(32\rho\sigma)^2}{\mu(\varepsilon\mu)^3}, \frac{(32\rho\sigma e^{3c/2})^3}{(\varepsilon\mu)^3}\right] \log\left(\frac{2(f(x_0) - f^\star)}{\varepsilon}\right),$$

ignoring the requirements $K \geq 2c$, $K \geq \frac{1}{2} \log\left(\frac{2(f(x_0) - f^\star)}{\varepsilon}\right)$, which holds in practice (and also follows from the assumption on $T$). Multiplying both sides of this expression by $BS$, using Assumption 4 that says that $\sigma^2 = \frac{\sigma_\star^2}{BS}$, and using the relation $T = KBS$, we obtain

$$T = \max\left\{\frac{128Le^{3c}LBS}{\varepsilon\mu^2}, \frac{32e^{3c/2}\rho\sigma_\star\sqrt{BS}}{\varepsilon\mu}, \frac{128Le^{6c}(32\rho\sigma_\star)^2}{\mu(\varepsilon\mu)^3}, \frac{(32\rho\sigma_\star e^{3c/2})^3}{(\varepsilon\mu)^3\sqrt{BS}}\right\} \log\left(\frac{2(f(x_0) - f^\star)}{\varepsilon}\right).$$

Since the token budget $T$ is fixed in the experiments, the expression above says that we cannot achieve an arbitrary optimization error $\varepsilon$:

$$\varepsilon = \max\left\{\frac{128e^{3c}LBS}{T\mu^2}, \frac{32e^{3c/2}\rho\sigma_\star\sqrt{BS}}{T\mu}, \left(\frac{128Le^{6c}(32\rho\sigma_\star)^2}{\mu^4 T}\right)^{1/3},\right.$$

$$\left.\left(\frac{(32\rho\sigma_\star e^{3c/2})^3}{T\mu^3\sqrt{BS}}\right)^{1/3}\right\} \log\left(\frac{2(f(x_0) - f^\star)}{\varepsilon}\right).$$

Now we compare the second and fourth terms in the expression above. We note that the second term is larger *iff*

$$\frac{32e^{3c/2}\rho\sigma_\star\sqrt{BS}}{T\mu} \geq \frac{32\rho\sigma_\star e^{3c/2}}{T^{1/3}\mu(BS)^{1/6}} \iff (BS)^{2/3} \geq T^{2/3} \iff BS \geq T. \tag{34}$$

In other words, the second term is smaller than or equal to the fourth term. Therefore, it can be ignored in the maximum. This finalizes the proof. $\square$

**Remark 4.** Our work is based on the convergence guarantees under the $\mu$-KL condition following prior work (Schaipp et al., 2025; Islamov et al., 2024; Tran et al., 2024; Guille-Escuret et al., 2024) that provides evidence that the loss landscape of neural networks exhibits a convex-like structure. However, it is possible to extend the results to a standard non-convex setting under the smoothness assumption only. In such a case, one can consider Unconstrained SCG (Algorithm 2) and the convergence metric changes from the function sub-optimality to a dual gradient norm, i.e., $\min_{k=0,1,...,K-1} \mathbb{E}[\|\nabla f(x_k)\|_*]$ or $\mathbb{E}[\|\nabla f(\overline{x}_k)\|_*]$ with $\overline{x}_k$ being selected uniformly at random from $\{x_0, x_1, \ldots, x_{K-1}\}$; see (Pethick et al., 2025a, Theorem 5.5) and (Kovalev, 2025, Corollary 2).

---

**Algorithm 2** Unconstrained Stochastic Conditional Gradient (uSCG)

---

**Input:** $x_0, m_0 \in \mathcal{X}$, parameters $\alpha, \eta > 0$
**for** $k = 0, \ldots, K - 1$ **do**
    sample $\xi_k \sim \mathcal{D}$
    compute $m_{k+1} = (1 - \alpha)m_k + \alpha g(x_k; \xi_k)$
    compute $d_{k+1} = \arg\min_{d \in \mathcal{X}} \langle m_{k+1}, d \rangle$ s.t. $\|d\| \leq 1$
    compute $x_{k+1} = x_k + \eta d_{k+1}$
**end for**

---

Under the setup of (Kovalev, 2025, Corollary 2) and a fixed token budget $T$, we achieve the optimization error

$$
\begin{aligned}
\varepsilon &= \mathcal{O}\left( \max\left\{ \frac{\sqrt{L\Delta BS}}{\sqrt{T}}, \frac{(L\Delta)^{1/4}\sqrt{\rho\sigma_\star}}{T^{1/4}}, \frac{\rho\sigma_\star\sqrt{BS}}{T}, \frac{\rho\sigma_\star}{T^{1/3}(BS)^{1/6}} \right\} \right) \\
&= \mathcal{O}\left( \max\left\{ \frac{\sqrt{L\Delta BS}}{\sqrt{T}}, \frac{(L\Delta)^{1/4}\sqrt{\rho\sigma_\star}}{T^{1/4}}, \frac{\rho\sigma_\star}{T^{1/3}(BS)^{1/6}} \right\} \right),
\end{aligned} \tag{35}
$$

where we used $\Delta = f(x_0) - f^\star$ and $T \geq BS$. We observe that the third term in (35) is identical to the third term in (3) (up to constant $\mu$, which is expected due to the change of convergence metric), while the first two are different. Besides, the middle term is also batch size and sequence length independent, but has a power $T^{1/4}$ instead of $T^{1/3}$ as in (3). Following the approach of Section 5, i.e., choosing $B$ and $S$ in the intersection of the first two terms in (36), we derive the scaling rules similar to (9)

$$
B_1 S_1 = B_0 S_0 \sqrt{\frac{D_1}{D_0} \frac{\rho_1^2}{\rho_0^2} \frac{L_0}{L_1}}, \tag{36}
$$

assuming that parameters $\Delta$ and $\sigma_\star$ are independent of the model size. This approach is similar to (15) up to problem-dependent constants $\rho$ and $L$.

