# OpenReview forum: "On the Role of Batch Size in Stochastic Conditional Gradient Methods"
_ICML.cc/2026/Conference — ICML 2026 regular_

### Official Review · Reviewer_pQhB · 2026-03-07

**Soundness:** 2
**Presentation:** 2
**Significance:** 3
**Originality:** 3
**Overall Recommendation:** 3
**Confidence:** 4

**Summary:**

This paper provides a theoretical analysis for the relationship of the batch size, learning rates and stochastic noise for the conditional gradient algorithms. Under the limited total token, there is a trade-off between the batch size and the sequence based on the convergence bounds. Based on the theoretical results, the paper proposes a two-stage strategy to tune these hyperparameters and provides empirical results to verify its effectness.

**Compliance With Llm Reviewing Policy:**

Affirmed.

**Key Questions For Authors:**

(a). From Table 2, it seems that the optimal selection is $B=512,S=1024, \beta= 3.6\times 10^{-4}$. Why the baseline uses $B=256, S=2048$?

(b). If the data distributions of $T_0$ and $T_1$ are different, which could happen in practical cases, does the two-stage stragety still work? From my view, since this stragety relies on problem parameters, it may be sensitive to the data distribution.

(c). Works like [1] use the fitted power function to capture the relationship of $B,S,T$ and obtain the so-called "critical batch-size", what's the advantage of the two-stage stragety over these methods?

[1]. An Empirical Model of Large-Batch Training, OpenAI.

**Strengths And Weaknesses:**

**Strengths**

(a). This paper provides some interesting results for the trade-off between the batch-size and the sequence length given the limited total tokens. Using the convergence upper bound, the authors derive the relationship between these hyperparameters (including the learning rate).

(b). Based on the theoretical results, they propose a novel two-stage training stragety, considering the case where the tokens are limited in Stage one and a small model is trained. In Stage two with more tokens and a larger model, the stragety can rely on the theoretical results to transfer the hyper-parameters in Stage one.

**Weaknesses**

(a). A key assumption for the two-stage stragety is that the problem-parameters such as $L$ and $\mu$ remain roughly same for small and large models. Although Table 3 provides some evidence to verify this, it's not very convincing from my view. First, the estimations are derived from only three sizes of models. Second, since the llm models are much more complicated and therefore usually not statisfy the $L$-smoothness or $mu$-KL condition. The parameters estimation in Table 3 could be local instead of global. Third, the estimations of problem parameters are heavily dependent on the data distribution, only using the results from one dataset (Fineweb) may be not so convincing.

(b). Since the stragety heavily relies on the estimation of problem-parameters, it's not so robust and requires accuracy for these estimations.  It may be better to provide some results to show that the two-stragety can be robust even the estimation for problem-parameters are not so accurate.

(c). The two-stage stragety relies on Eq. (5) and Eq. (6). These formulas indicate the requirement for $B \times S$. Are there any further requirements for $B$ or $S$? Why the experiments set $S=1024$ and $B=256$ instead of $S=2048$ and $B=128$?

---

> ### Author Rebuttal · Authors · 2026-03-30
>
> We thank the reviewer for the thoughtful comments and for highlighting the interaction between $B,S,\beta$ and the practical relevance of the two-stage strategy. We address the concerns below.
>
> ### Weaknesses
>
> W1: We would like to clarify that Table 3 reports **fitted scaling laws for the problem-dependent constants**, not estimates based on only three models. The fitting is performed using a significantly larger set of configurations (Tables 5–7), which include variations in depth and embedding size.
>
> More importantly, our approach does **not rely on precise estimation of $(L,\mu,\rho)$**, but rather on their **relative scaling behavior across models**. This distinction is crucial. The BST rule depends on the *scaling regime* (e.g., how quantities change with model size and token budget), not on exact numerical values.
>
> Empirically, these quantities stabilize after an initial transient (see **[[L](https://anonymous.4open.science/r/Figures-971E/L_dynamics.png)]** and **[[$\rho$](https://anonymous.4open.science/r/Figures-971E/rho_dynamics.png)]**), which makes them suitable for estimating scaling behavior. Furthermore, since $B$ and $S$ are chosen from a discrete set (typically powers of two), the final configuration is inherently robust to moderate estimation error.
>
> Finally, the validity of these estimates is **empirically confirmed** by successful transfer to the 1B model, where BST-based configurations significantly outperform the $\mu$P baseline.
>
> ---
>
> W2: We agree that robustness is important, but our results indicate that the method is **not sensitive to precise parameter estimation**.
>
> There are three reasons for this:
> 1. The BST rule determines the **scaling regime** (e.g., $BS \sim T^{2/3}$), not exact values.
> 2. The final $(B,S)$ choices are discretized (powers of two), which naturally absorbs estimation noise.
> 3. The empirical results (Tables 1–2 and 8–11) show **consistent performance improvements**, indicating that the method is stable in practice.
>
> In addition, our estimates are consistent with independently observed scaling behavior for SCG-type methods in [1], providing further evidence that the identified scaling is not fragile.
>
> [1] Filatov, Oleg, et al. "Optimal scaling needs optimal norm." arXiv preprint arXiv:2510.03871 (2025).
>
> ---
>
> W3: We agree that the theory constrains the product $BS$. This is by design: under a fixed token budget $T = KBS$, the optimization dynamics depend on $BS$.
>
> The decomposition into $(B,S)$ is then determined by practical considerations. In particular:
> - $S$ affects model capability,
> - $B$ affects optimization efficiency.
>
> Our experiments (Tables 1–2) show that, even at similar $BS$, different decompositions can yield different performance. Therefore, we:
> (i) use the theory to determine the appropriate scale of $BS$, and
> (ii) select $(B,S)$ empirically.
>
> For the specific example raised by the reviewer, $B=256,S=1024$ outperforms $B=128,S=2048$ in our sweeps, which is why it is chosen.
>
> ### Questions
>
> Q1: According to Table 1, the configuration $B=256,S=1024,\beta=3.6\cdot10^{-4}$ achieves lower validation loss (3.275 vs 3.321). We therefore select the empirically best-performing configuration.
>
> ---
>
> Q2: The theoretical analysis is stated for a fixed objective function (i.e., fixed data distribution), which is standard in optimization theory.
>
> In practice, scaling laws in deep learning are typically observed to be **stable across models trained on the same dataset**, which is the setting we consider. In this regime, the relevant quantities $(L,\mu,\rho)$ reflect properties of the loss landscape induced by the data and architecture, and their **relative scaling across models** is the key factor for transfer.
>
> We agree that studying robustness under distribution shift is an interesting direction, but it does not affect the validity of the scaling law in the setting considered in this paper.
>
> ---
>
> Q3: The key difference is that our rule is derived from **optimization bounds on the loss (suboptimality)**, rather than from fitted relationships or gradient-noise-based surrogates.
>
> As a result:
> - BST provides a **token-budget-aware scaling rule**, explaining how $B,S,\beta$ should change with $T$,
> - rather than identifying a single critical batch size for a fixed setup.
>
> While both approaches can produce similar qualitative observations (e.g., the existence of a critical scale), our result provides a **principled derivation tied directly to the optimization objective**, and is validated on full transformer pre-training experiments.

---

> > ### Author Rebuttal · Reviewer_pQhB · 2026-04-07
> >
> > Thanks a lot for your reply. For W1, I think that some more evidence is required. For W3, the authors claim that even under the same BS, the different B and S may lead to very different results. From this view, if using the scaling law from the paper, we can only obtain a BS and should further use methods like grid search to determine the specific selection of B and S, which requires more cost.
> >
> > Also, I am confused that if B=256 is selected from Table 1 through a grid search of small model, why the larger model (1B) baseline is selected with B=512? is it also use a grid search? Since using the scaling law could only obtain a BS for the larger model. If B = 512 is just selected through the grid search in 1B model, then it may cost a lot as I said in the first point.

---

> > > ### Author Response · Authors · 2026-04-07
> > >
> > > We thank the reviewer for these follow-up questions. We realize that the paper did not make the **calibration-vs-transfer workflow** sufficiently explicit, and this is likely the source of the confusion. We will revise the presentation to make this point clearer.
> > >
> > > For **W1**, the reviewer’s concern would indeed be valid **if** our method required re-running a grid search over $(B,S)$ at each larger scale. This is **not** what we do. The purpose of BST is precisely to **avoid repeating the expensive large-scale search**.
> > >
> > > The workflow is:
> > > 1. use the small model to identify a good practical decomposition $(B,S)$,
> > > 2. use BST to determine how the corresponding tokenized batch size $BS$ should scale with token budget,
> > > 3. transfer this prescription to the larger model by updating $B$ accordingly while keeping the chosen $S$ fixed unless there is a separate reason to change it.
> > >
> > > So the large-scale step is a **transfer step**, not a fresh grid search. This is why the method remains practical.
> > >
> > > This is also where BST is explicitly distinct from $\mu$P-style transfer. $\mu$P assumes the same batch size across scales, so it does not address how batch size itself should be adjusted when moving to a larger model. BST does: it allows us to tune the small model once and then **predict the appropriate large-model batch size** through the scaling rule, rather than retuning it at large scale.
> > >
> > > For **W3**, we agree that the current theory identifies the correct scale of $BS$, not a unique decomposition into $B$ and $S$. This is intentional. In practice, $S$ is not just an optimization parameter: it is tied to capability and deployment requirements (e.g., context length), whereas $B$ primarily controls optimization efficiency and noise. Therefore, the intended use is not to grid-search $(B,S)$ repeatedly at every scale, but to choose a reasonable $S$ once using a small-model sweep and practical constraints, and then use BST to scale $BS$ with budget.
> > >
> > > This also clarifies an additional practical use case of BST. If one wants to **increase the context length during training** (i.e., increase $S$ for capability reasons), BST indicates that one should **adapt the batch size $B$ accordingly** so that the tokenized batch scale remains in the correct regime. In this sense, BST is not only a transfer rule across budgets, but also a prescription for how to retain computational efficiency when changing sequence length during training.
> > >
> > > This also resolves the question about the 1B baseline. The larger-model baseline with $B=512$ was **not** selected by a 1B grid search. The base configuration is calibrated on the small model: from Tables 1--2, we select $B=256, S=1024$ as the preferred small-model configuration. When moving to the larger 1B budget, BST prescribes increasing the product $BS$ four times; we decided to use $B=512$ and $S=2048$ at 1B scale. In other words, the 1B choice is obtained by **transferring the small-model calibration through the BST rule**, not by re-running the same expensive search at larger scale.
> > >
> > > **Additionally, we include experiments with more baselines**: two where $B$ and $S$ are increased to 512 and 2048 (either from the start or after restart), and two where $B$ is increased to 1024 (again, either from the start or after restart). We refer the reviewer to the results in the provided figure. We observe that BST-based baselines with larger $BS$ products substantially outperform the $\mu$P baseline with $B=256$ and $S=1024$, which was derived from a smaller 124M model.
> > >
> > > The point of BST is therefore not to eliminate the initial small-scale calibration, but to **avoid repeating it at every larger scale**. We will revise the paper to make this workflow explicit.
> > >
> > > We also note that the transferred prescription is empirically supported: the constants used for transfer exhibit stable behavior after an initial transient; see **$L$ (https://anonymous.4open.science/r/Figures-971E/L_dynamics.png)** and **$\rho$ (https://anonymous.4open.science/r/Figures-971E/rho_dynamics.png)**. In addition, the predicted Frank--Wolfe stepsizes match the empirically best values in **$\beta$ sweeps (https://anonymous.4open.science/r/Figures-971E/betas.png)**, which further supports that the transferred prescription is meaningful rather than fragile.

---

### Official Review · Reviewer_TnA6 · 2026-03-11

**Soundness:** 3
**Presentation:** 3
**Significance:** 2
**Originality:** 3
**Overall Recommendation:** 4
**Confidence:** 4

**Summary:**

The paper performs a analysis on how the batch size, sequence length scale with token budget for Frank–Wolfe–type conditional gradient algorithms. They pin down the notion of the critical batch size beyond which increasing the batch size gives diminishing returns and may even degrade the performance. The authors also propose a batch size scheduling algorithm which gives matching rates in comparison to the setting where this new horizon in know in advance.

**Compliance With Llm Reviewing Policy:**

Affirmed.

**Final Justification:**

The rebuttal and the additional references provided by the authors addressed my questions and concerns and hence I have decidd to increase my score.

**Key Questions For Authors:**

* Have the optimizer momentum parameter, the EMA coefficient (Beta), and the learning rate been tuned for every batch size/sequence length pair?
* For the batch size scheduling experiments, how has the warmup been set?
* In practice, for most popular optimizers, the critical batch size scales as sqrt(token budget), whereas here it scales as T^2/3. Have the authors verified this empirically? Does the theory hold in practice?
* Why hasn’t Nano-GPT been trained for Chinchilla optimal tokens (for Table 2 results, whereas, for figure 2 they have been trained for Chinchilla optimal tokens)?
* The notion of a critical batch size in theory usually comes up when balancing the bias and the variance of the loss, how does the proposed theory relate to this notion of the critical batch size?
* Does the proposed batch size scheduling algorithm give any benefit if the token budget is already known beforehand, like in the case of Seesaw (Meterez et al., 2025).
* Citations: Zhang et al., (2024): How Does Critical Batch Size Scale in Pre-training?
  * Meterez at el., (2025): Seesaw: Accelerating Training by Balancing Learning Rate and Batch Size Scheduling.
  * Meterez at el., (2026): Anytime Pretraining: Horizon-Free Learning-Rate Schedules with Weight Averaging.
  * WSD: Minicpm: Unveiling the potential of small language models with scalable training strategies.

**Limitations:**

* The scaling law proposed in the paper hasn’t been empirically tested.
* The proposed batch size scheduling method hasn’t been compared to existing continual pretraining methods.

**Strengths And Weaknesses:**

Strengths:
* The theoretical arguments made in the paper are convincing.
* The results showcase the improvements brought by the batch size and sequence length scheduling scheme.

Weakness:
* For batch size scheduling the paper does not compare with continual pretraining algorithms such as WSD which can obtain minimax optimal rates (up to log factors) for overparameterized linear regression with polynomially decaying spectrums, as shown in Meterez et al., (2026).
* Zhang et al., (2024) show that the sequence length vs batch size (i.e., number of training sequences in a batch) curves are flat, i.e., the scaling is always in the total number of tokens with a single batch. Why model B and S separately?
* The models in Table 2 have not been trained for Chinchilla optimal tokens making the comparisons unfair.

---

> ### Author Rebuttal · Authors · 2026-03-30
>
> Thank you for thoughtful comments and for recognizing the theoretical nature of our contribution.
>
> ### Weaknesses
>
> W1: Connections to continual pretraining and schedule design (WSD) are relevant, but our focus differs in two key aspects: 1) we study **full transformer pre-training**, not simplified models; 2) our goal isn't to design a specific schedule with optimality guarantees in a particular model class, but to derive a **token-budget-aware scaling law** that explains how batch size and related parameters should scale in realistic training. Thus, our contribution is complementary: **scaling law derived from optimization bounds** that can guide practical training.
>
> W2: Indeed, many results suggest that performance depends primarily on token budget T. But, in practice, B and S aren't interchangeable: S affects model capability (context length and expressivity); B affects optimization (variance, hardware efficiency). Our empirical results (Tab 2) show that, even at fixed BS, different (B,S) choices lead to different performance, indicating that the decomposition matters in practice. Our theory identifies the **scale of BS** as the key quantity under a fixed token budget, while the choice of how to split it into (B,S) remains a practical design decision.
>
> W3: Training at Chinchilla-optimum is desirable, but our experiments are constrained by available compute. The question we study is **comparative**: for a given T, how should $B,S,\beta$ be chosen? This question remains meaningful even away from the Chinchilla-optimum. We also refer the reviewer to Tab 8–11, where we report results at larger token budgets for 124M model. Additionally, in **[[$\beta$ sweeps](https://anonymous.4open.science/r/Figures-971E/betas.png)]**, we show that empirically optimal and predicted FW stepsizes match up to statistical error, supporting the predictive power of BST rule.
>
> ### Questions
>
> Q1: We didn't initially perform full sweeps (e.g., momentum) for each configuration, as this would be computationally prohibitive. However, to assess sensitivity, we include sweeps for momentum $\alpha$ at B=256,S=1024,$\beta$=36e-5: **[[URL](https://anonymous.4open.science/r/Figures-971E/alphas.png)]**, where the empirical optimal value is around 0.09, and the performance at $\alpha=0.1$ is within statistical error
>
> Q2: We don't use warmup in our experiments, following prior works on SCG methods (Filatov et al, Optimal scaling needs optimal norm; Pethick et al, Training deep learning models with norm-constrained lmos)
>
> Q3: The main limitation is computational. Training 1B models at Chinchilla-optimal scale isn't feasible within our available resources. To mitigate this, we i) perform detailed sweeps at 124M scale across multiple token budgets, ii) use these to determine optimal configurations, and ii) transfer them to larger models. We also refer the reviewer to Tab 8–11 and additional sweeps above, showing consistent behavior across token budgets
>
> Q4: Our result is closely related in spirit but differs in formulation. Classical notions of critical batch size arise from balancing bias and variance via GNS, which is an auxiliary object towards the final goal. In our case, the trade-off emerges directly from the **optimization bound on suboptimality** under a fixed T. Eq (2) contains competing terms: **deterministic term**, which worsens with $BS/T$ and **stochastic term**, which reflects gradient noise. BST rule is obtained by balancing these terms, leading to scaling $BS\sim T^{2/3}. $ Thus, our notion of “critical scale” corresponds to the transition between deterministic and stochastic regimes, but is derived from **optimization error rather than gradient noise**
>
> Q5: We believe Seesaw and BST address related but different questions. Seesaw is designed to match the performance of cosine annealing using piecewise-constant LR and batch-size schedules under fixed T. In this case, the batch-size evolution is coupled to the LR schedule. By contrast, BST is a **token-budget-aware scaling rule**: it explains how to choose and adapt B, S, and $\beta$ as T changes. This difference matters in practice: under schedulers such as stable-decay (used in our experiments) the batch size wouldn't necessarily evolve in the same way as under Seesaw-style coupling to cosine annealing. BST instead provides a direct prescription for how parameters should change when T changes
>
> BST can also be used in a fixed-budget multi-stage setting. If total $T$ is split into stages $T_1,...,T_S$, one can first identify good parameters for the initial stage $T_1$, and then update $(B,S,\beta)$ at stage $s$ according to BST rule using the cumulative budget $\sum_{i=1}^s T_i$. Our two-stage experiments correspond to this setup, where $T_1$ is the smaller-budget stage and $T_1+T_2$ is the final budget. So even when the final budget is known in advance, BST is useful whenever training is done in stages, or the effective operating regime changes over the course of training

---

> > ### Author Rebuttal · Reviewer_TnA6 · 2026-04-02
> >
> > I thank the authors for a descriptive response.
> > Has the proposed scaling (BS = T^2/3) been validated through scaling laws?

---

> > > ### Author Response · Authors · 2026-04-02
> > >
> > > We would like to clarify that our goal is different from classical *fit-first* empirical scaling-law work. Rather than fitting an exponent from large sweeps, we **derive** the scaling law
> > > $$
> > > BS \sim T^{2/3}
> > > $$
> > > from optimization bounds, and then validate that the resulting prescription is empirically useful.
> > >
> > > This distinction is important because empirical scaling laws primarily provide **predictability within a given experimental setup**, but not necessarily **optimality for the underlying optimization problem**. Recent work such as *Gemstones* further shows that innocuous changes in architecture, hyperparameters, or checkpoint selection can materially change the resulting empirical scaling prescription. In that sense, our result plays a different role: it provides a **theoretical certificate for how one should scale $BS$ with token budget**, rather than relying solely on post hoc fitting.
> > >
> > > Within this framework, our paper already provides several forms of empirical validation:
> > > - the BST rule predicts practically useful configurations as the token budget changes,
> > > - the predicted Frank--Wolfe stepsizes match the empirically optimal values (see **[[$\beta$ sweeps](https://anonymous.4open.science/r/Figures-971E/betas.png)]**),
> > > - and transfer from smaller to larger models improves performance when following BST-derived configurations.
> > >
> > > At the same time, we do not want to suggest that empirical evidence is unimportant. Our prediction is also supported by **independent large-scale empirical results** in Filatov et al. [1], where optimal learning-rate/batch-size scaling for SCG-type methods is measured across substantially larger models and datasets. Their setup is not fully aligned with ours, so one should not expect exact quantitative agreement; however, the scaling trends they observe are consistent with our rule and provide additional support that BST captures a real large-scale phenomenon.
> > >
> > > Finally, the same $BS \sim T^{2/3}$ scaling arises from an independent derivation for SGD under the same assumptions, suggesting that this is a more general phenomenon and not an artifact of SCG alone (We kindly refer to our response Q1 to Reviewer Tbqk, where the derivations for SGD are provided.).
> > >
> > > [1] Filatov, Oleg, et al. “Optimal scaling needs optimal norm.” arXiv preprint arXiv:2510.03871 (2025).

---

### Official Review · Reviewer_ohAX · 2026-03-12

**Soundness:** 3
**Presentation:** 2
**Significance:** 2
**Originality:** 3
**Overall Recommendation:** 4
**Confidence:** 4

**Summary:**

This paper investigates the impact of batch size on momentum-based Stochastic Conditional Gradient (SCG) methods under a $\mu$-Kurdyka-Lojasiewicz ($\mu$-KL) condition, specifically within a fixed token budget. The authors derive a "BST scaling rule" ($BS \approx T^{2/3}$) and identify three scaling regimes (noise-dominated, independent, and iteration-starved) where optimization error behaves non-monotonically with respect to batch size. Leveraging these insights, they propose an adaptive scheduling strategy that increases batch size and sequence length during training, complementing local stability frameworks like $\mu$P by addressing global optimization efficiency, with empirical validation on NanoGPT models.

**Compliance With Llm Reviewing Policy:**

Affirmed.

**Final Justification:**

The authors have provided a thorough response to my initial concerns regarding the perceived inconsistencies between the theoretical framework and the experimental conclusions. The rebuttal clarified that these discrepancies primarily stemmed from the specific presentation style and the choice of visualization ranges in the plots, rather than fundamental flaws in the logic. Furthermore, the newly included 3-stage restart experiments provide solid empirical evidence that aligns with their theoretical predictions and strengthens the overall paper. I am satisfied with these clarifications and the additional work performed. Consequently, I am raising my rating to 4.

**Key Questions For Authors:**

1. The validation experiments in this paper are restricted to relatively small models with 124M, 775M, and 1B parameters. Current mainstream large language models typically operate at scales of 7B or even tens of billions of parameters. It remains highly questionable whether the empirical power-law fits proposed to guide parameter transfer (e.g., how the constants $L, \mu, \rho$ scale with the number of model layers) remain valid at much larger, realistic scales.
2. When estimating the smoothness constant $L$ and the norm equivalence constant $\rho$, the authors employ a simple heuristic of taking the "average over the last 100 iterations." In reality, the curvature of the loss function fluctuates drastically throughout the entire pre-training cycle. Approximating a global property using such a highly localized metric lacks rigorous justification. Are there any established works in the literature that use this exact method of representing global constants with $n$ local iterations?
3. In Figure 3 (1B model validation loss), the authors demonstrate that the Restarted SCG strategy (the red line, transitioning from a small batch to a large batch) ultimately outperforms the baseline that consistently uses the globally optimal large batch (the blue line). This empirical observation contradicts the core theoretical narrative of the paper. The proposed BST scaling law merely derives the optimal constant $BS$ under a static total budget $T$. How can the theoretical framework explain why "multi-stage dynamic scheduling" yields better performance than the strictly derived "static globally optimal hyperparameters"?
4. Following the authors' theoretical logic, if the training is evaluated at an intermediate stage, the conclusion should align with the scenario where the budget is $T = T/N$; that is, a smaller $BS$ should yield better performance, whereas a larger $BS$ becomes optimal only at the end of the full training cycle. Extending this logic, the earlier the training stage, the better a smaller $BS$ should perform. This crossing phenomenon is completely absent in Figure 3, representing a direct contradiction to the theoretical derivations. If this absence is merely due to the chosen $BS$ differences not being large enough, the authors should provide an additional experimental plot clearly exhibiting this crossing trend to properly validate their theory.

**Limitations:**

see weakness

**Strengths And Weaknesses:**

Strengths
1. The problem addressed is practically significant. Traditional optimization analysis typically focuses on convergence rates as the number of iterations approaches infinity. In contrast, this paper incorporates the realistic constraint of finite Token budgets in Large Language Model (LLM) training scenarios, successfully transforming iteration complexity into Token complexity.
2. The theoretical assumptions are reasonable and relatively weaker than those in existing literature, making them more aligned with the actual characteristics of large models.
3. The theoretical proofs are solid, successfully deriving valid conclusions regarding the three distinct training regimes: noise-dominated, independent, and iteration-starved.
4. In addition to theoretical findings, the authors propose a theoretically-grounded hyperparameter transfer strategy. They demonstrate that directly and statically applying the currently popular $\mu$P framework to large models yields suboptimal results, thereby extending the paper's theoretical contributions from mere existence to practical utility.
5. Moving beyond ideal settings, the paper investigates practical scenarios such as "delayed data arrival" or dynamically growing computational budgets, and proposes the "Restarted Scion" strategy as a potential solution to these challenges.

Weaknesses
1. Both the theoretical derivation and experimental evaluation in the paper are strictly based on Scion, a specific stochastic conditional gradient (SCG) method. However, the vast majority of current language models are pre-trained using adaptive optimizers such as AdamW. The authors neither discuss nor prove whether this token budget-based batch size scaling law can be generalized to standard optimizers outside the conditional gradient family, nor do they demonstrate the general applicability of this method in large language model training.
2. The derivation of Theorem 1 almost entirely follows the classical analysis frameworks of existing SCG or Frank-Wolfe algorithms. As acknowledged by the authors, this bound is highly similar to results derived under the star-convexity assumption in existing literature. Although introducing the $\mu$-KL condition has practical value, it represents merely an incremental improvement in the context of pure mathematical optimization theory, which weakens the theoretical contribution of the authors.
3. Theorem 1 only proves the error convergence upper bound for a single model given a suitable set of constants (L,μ,ρ). However, the strategy for "cross-model hyperparameter transfer" heavily relies on empirical power-law fitting to extrapolate how these constants scale with model size (D), and the accuracy of such extrapolation is fundamentally unreliable. For a predominantly theory-based study, this empirical approach undermines the mathematical rigor of the scaling law. If possible, the authors should supplement this with more rigorous theoretical derivations or provide stronger justifications for the validity of the power-law fitting.
4. The theoretical analysis only constrains tthe product of $B$ and $S$, without decoupling them at the mathematical level. That is, although the original text claims that "This section establishes convergence guarantees for Algorithm 1, guiding how to choose the batch size $B$, sequence length $S$, and step size $\beta$ under a fixed token budget $T$," the mathematical theory proposed by the authors does not actually achieve this decoupling of B and S in practice.

---

> ### Author Rebuttal · Authors · 2026-03-30
>
> We thank the reviewer for the feedback and for recognizing the practical relevance of fixed token budgets and our scaling rules. We address the main concerns below.
>
> ### Weaknesses
>
> W1: We kindly refer to Q1 in response to Reviewer Tbqk for details.
>
> - - -
>
> W2: We respectfully disagree with this claim. The contribution isn't only deriving bounds, but **translating them into a token-budget-aware scaling law**. Our analysis operates under $\mu$-KL cond., derives a fixed-token-budget error bound, reveals a U-shaped dependence on $BS$, and yields a best achievable error of order $T^{-1/3}$, leading to the rule $BS\sim T^{2/3},$ which links optimization dynamics with the token budget. Prior work identified **critical batch sizes** or proposed **empirical scaling laws**; none derived an explicit **token-budget-dependent batch scaling law from optimization bounds**. Technically, $\mu$-KL analysis isn't a cosmetic change. In our proof, descent must be established by controlling terms such as $\langle\nabla f(x_k),x_k\rangle$ indirectly via norm bounds on $\\|x_k\\|$, which isn't required under star-convexity. Unlike star-convexity, $\mu$-KL condition can be **empirically assessed without access to $x^*$** (Fig 1). Our theoretical prediction is supported by recent work [1], thas show scaling behaviors consistent with our rule across training regimes.
>
> [1] Filatov et al, Optimal scaling needs optimal norm, arXiv:2510.03871, 2025
>
> - - -
>
> W3: Note that the **theoretical result and the transfer procedure are distinct components** of the work. BST rule is derived from the convergence analysis and does **not depend on power-law fitting**. Power-law fitting is used as a **practical mechanism** to estimate how $L,\mu,\rho$ vary with model size for transfer. The fitting bridges **per-model theory** and **cross-model application**. Empirically, we observe that the constants stabilize after an initial phase and follow smooth trends across model sizes (see **[[L](https://anonymous.4open.science/r/Figures-971E/L_dynamics.png)]** and **[[$\rho$](https://anonymous.4open.science/r/Figures-971E/rho_dynamics.png)]**). Also, since B and S are selected as powers of 2, the transfer procedure doesn't require highly precise estimates of the constants, but identification of the relevant scaling regime. We agree that a fully theoretical characterization of how $(L,\mu,\rho)$ scale with model size is an important open problem, and we will clarify this limitation in the revision.
>
> - - -
>
> W4: The theoretical analysis identifies the **product BS** as the key quantity under a fixed token budget. The roles of B and S are separated at the practical level: S primarily affects model capability (e.g., context length); B primarily affects optimization efficiency and gradient variance. In our experiments, we therefore: i) use the theory to determine the appropriate scale of BS, and ii) select the specific (B,S) decomposition based on empirical performance. We will revise the wording to clearly distinguish between these two levels.
>
> ### Questions
>
> Q1: We agree that extending the empirical validation to larger scales is important. Our experiments are constrained by available compute, and we will clarify this limitation more explicitly. Our goal is to demonstrate that the predicted scaling structure is already observable across multiple model sizes and is sufficiently stable to support transfer. We view larger-scale validation as an important next step.
>
> ---
>
> Q2: We agree that $L$ and $\rho$ are global quantities in theory, while any empirical estimate in neural training is necessarily local or trajectory-dependent. Our use of a running average over the last 100 iters isn't intended as an exact estimator of a global constant, but as a **stabilized proxy for the regime where training spends most of its time after the initial transient**. Empirically, these estimates become significantly more stable after early training, which is why they are useful for transfer; see **[L]** and **[$\rho$]** above. We will clarify this interpretation in the revision.
>
> ---
>
> Q3: We found that the original comparison was confounded by varying validation sequence length. We therefore updated Fig 3 by fixing the val seq. len. to 1024 throughout, making the comparison fairer: **[[URL](https://anonymous.4open.science/r/Figures-971E/1B_val_train_loss.png)]**. The fixed large-batch-sequence-length baseline slightly outperforms the restarted version both in train and val loss, which is consistent with the fixed-budget theory. Moreover, both are significantly better than $\mu$P baseline. This should be viewed as the key point of the experiment: a token-budget-aware BST rule provides a much better practical prescription in this setting.
>
> ---
>
> Q4: In the same **[URL]**: the smaller-BS baseline improves faster early in training, while the larger BS baseline overtakes later. This is consistent with our fixed-budget analysis, and the updated figure now makes this behavior explicit.

---

> > ### Author Rebuttal · Reviewer_ohAX · 2026-04-02
> >
> > I appreciate the authors' detailed response and accept the explanations regarding novelty and the experimental figures. However, the response to Q4 does not fully resolve my concern.
> >
> > Following the exact logic of your proposed theoretical framework: if a strictly smaller total budget $T_{small}$ dictates a smaller optimal $BS_{small}$, then at any intermediate or early stage of a training run with a large budget $T_{big}$, the consumed token count $T_{current}$ is functionally equivalent to a smaller budget $T_{small}$. Therefore, a natural logical corollary of your theory seems to be that progressively smaller batch sizes should perform better at progressively earlier stages, and the optimal batch size should monotonically increase as the consumed token budget grows. Consequently, the training trajectories of varying $BS$ configurations should sequentially overtake one another.
> > If my understanding is correct, it would strengthen the paper to include at least 1-2 additional intermediate $BS$ configurations in the experiments to continuously validate this crossing phenomenon. More importantly, validating this would imply that the ultimate conclusion of the paper might need to shift from proposing a "static globally optimal hyperparameter" to explicitly advocating for a "dynamic, monotonically increasing batch size schedule."
> >
> > If my interpretation is incorrect, I would greatly appreciate a theoretical clarification on why this corollary does not hold. If the authors can effectively address this critical issue, I will consider raising my score.

---

> > > ### Author Response · Authors · 2026-04-02
> > >
> > > We thank the reviewer for this insightful follow-up. We believe the intuition behind the question is closely related to an **anytime / horizon-free** interpretation of our result, and this is indeed the right conceptual lens.
> > >
> > > Our current theorem solves a **fixed-horizon optimization problem**: for a target token budget $T$, it identifies the static scale of $(B,S,\beta)$ that minimizes the terminal error after spending that full budget. In this sense, it is analogous to a horizon-dependent choice such as
> > > $$
> > > \beta_T = O(T^{-1/3}).
> > > $$
> > >
> > > The reviewer’s proposed corollary corresponds to a different and strictly stronger problem: an **anytime policy** that remains near-optimal as the consumed budget grows. In that setting, one would naturally expect a dynamic rule of the form
> > > $$
> > > \beta_k = O(k^{-1/3}), \qquad B_kS_k = O(k^{2/3}),
> > > $$
> > > up to problem-dependent constants. We agree that this is a natural and theoretically interesting extension of our theory.
> > >
> > > However, this does **not** follow automatically from the present theorem. The reason is that fixed-horizon and anytime optimization are different variational problems. Our analysis optimizes a **terminal bound at time $T$** under static hyperparameters. By contrast, an anytime theorem would need to control a **nonstationary recursion** with time-varying batch size, sequence length, and stepsize, and would have to show that the resulting trajectory is near-optimal for all prefixes. A prefix of a run optimized for horizon $T$ is therefore not equivalent to an independently optimized run whose total horizon is that prefix.
> > >
> > > This distinction is also practically relevant here: our large-model “hero” training run is **not** a streaming / unknown-horizon setting. It is planned in advance with a fixed target budget and timeline. So the appropriate theoretical object for the current paper is the **fixed-horizon optimum**, not an online policy that must remain competitive at every intermediate budget.
> > >
> > > In particular, the current theory does **not** establish a theoretical improvement from a dynamic increasing-$BS$ schedule over the best static choice tuned for the terminal budget $T$. This is consistent with the general role of anytime policies in optimization: they are attractive because they remove dependence on the final horizon, but they do not typically improve on the best horizon-tuned policy when that horizon is already known.
> > >
> > > So we believe the correct interpretation is:
> > > - our theorem establishes the optimal **static** scaling for a given terminal budget $T$;
> > > - the reviewer’s monotone-schedule intuition points to a plausible **dynamic / anytime** extension;
> > > - but that extension requires an additional theorem and should not replace the main conclusion of the current paper.
> > >
> > > Empirically, under the corrected evaluation protocol with fixed validation sequence length 1024, we do observe the expected qualitative trend in **[[URL](https://anonymous.4open.science/r/Figures-971E/1B_val_train_loss.png)]**: smaller $BS$ helps more early, while larger $BS$ is better later. We agree that adding more intermediate $BS$ values would be an interesting empirical probe of this anytime interpretation. We will revise the paper to make explicit that our current result is a fixed-budget static theorem, while an increasing-BST anytime schedule is a natural extension suggested by the same balance of optimization and stochastic terms.
> > >
> > > ---
> > >
> > > # **Update:**
> > >
> > > Following the reviewer’s request, we additionally performed a 3-stage restart experiment. Concretely, we use batch size 256 before processing $T_1$=1.3B tokens, batch size 512 between $T_1$ and $T_2$=3.8B tokens, and batch size $1024$ thereafter. The intermediate switching point $T_2$ is chosen so that batch size 512 follows the BST rule. Importantly, this experiment remains **within our theoretical framework**: at each stage $s$, the batch size is set according to the **cumulative number of processed tokens**, in line with the staged BST strategy analyzed in the paper.
> > >
> > > We compare this 3-stage strategy against two 2-stage baselines: one where both batch size and sequence length are doubled at restart, and one where batch size is increased fourfold. The resulting curves are shown in **[[URL](https://anonymous.4open.science/r/Figures-971E/3phases.png)]**. We observe that the 3-stage strategy achieves the **best validation loss**, improving over the 2-stage baselines by about $0.01$ at the end of training. Meanwhile, the train loss shows a different trend, with the 3-stage strategy slightly worse.
> > >
> > > We view this as useful numerical evidence: in our setup, multiple restarts can improve validation performance, although our theory does not claim that more restart stages are always superior. Rather, BST prescribes the correct scaling at each stage based on the cumulative token budget. Extending the analysis to a genuine anytime setting with many restarts is a separate question that we leave for future work.

---

### Official Review · Reviewer_Tbqk · 2026-03-13

**Soundness:** 4
**Presentation:** 4
**Significance:** 4
**Originality:** 4
**Overall Recommendation:** 5
**Confidence:** 3

**Summary:**

This paper presents and analysis of how batch size, sequence length, and stepsize interact in stochastic conditional gradient methods (such as Scion), deriving a scaling rule that identifies three regimes: a noise-dominated regime where increasing batch size helps, an intermediate regime where error is independent of batch size, and a large-batch regime where performance degrades under a fixed token budget. Based on these insights, the authors derive rules for adapting batch size, sequence length, and learning rate during training, increases batch size and sequence length as the token budget grows. Experiments on NanoGPT models up to 1B parameters on the FineWeb dataset show that the adaptive restarted strategy outperforms both fixed small-batch and fixed large-batch baselines.

**Compliance With Llm Reviewing Policy:**

Affirmed.

**Final Justification:**

I maintain my score of recommending the acceptance of the paper.

**Key Questions For Authors:**

How would these results transfer to more basic optimziers, such as Adam or SGD?

**Limitations:**

Training models only up to 1B parameters in technically a limitation, since the findings are most interesting to apply in the (very) large-scale training regime, but due to compute constrains this is understandable

**Strengths And Weaknesses:**

Strengths
- Batch size scaling rules are currently of great interest, as they directly impact the number of GPUs to use in a data-parallel training environment and therefore also impact the training efficiency
- The theoretical findings from Sec. 3 and 4 are verified by practical experiments
- Insights into the adaptive batch size scaling are directly usable by practitioners

Weaknesses
- A bit more discussions on related work of optimal batch size scaling strategies recently proposed (if though only for the Adam optimizer) is missing, as well as how the proposed method connects to related work on adapting the batch size during the training based on measurements such as the Gradient Noise Scale (e.g. [1, 2])
- No downstream evaluations are considered

[1] https://arxiv.org/abs/2411.00999

[2] https://arxiv.org/abs/2505.23971

---

> ### Author Rebuttal · Authors · 2026-03-30
>
> We thank the reviewer for the positive assessment and for highlighting the practical relevance of batch-size scaling and our theoretically motivated scaling rules
>
> ### Weaknesses
>
> W1: We will expand the related-work section to better position our contribution relative to recent batch-size scaling and batch-adaptation methods. GNS-based approaches [1,2], following [3], identify a critical batch size through gradient noise measurements. These methods have been highly influential in practice and provide a useful variance-based perspective on large-batch training. Our perspective is complementary and operates at a different level. GNS methods focus on **characterizing gradient noise**—an auxiliary quantity—often through approximations of the local loss geometry. In contrast, the BST rule is derived directly from **optimization bounds on the loss (suboptimality)** under the $\mu$-KL condition
>
> This leads to a different type of insight: our analysis explicitly balances an **optimization (iteration) term** and a **stochastic (noise) term** under a fixed token budget $T$. As a result, we obtain a scaling rule that aims to make stochastic training **as close as possible to its deterministic counterpart**, while accounting for how noise scales with $BS$. Moreover, Eq. (2) predicts a **U-shaped dependence on $BS$** and a best achievable error scaling of order $T^{-1/3}$, from which the BST rule follows. We will clarify these connections more explicitly in the revision
>
> [3] McCandlish et al., An empirical model of large-batch training,arXiv:1812.06162,2018
>
>
> W2: We agree that downstream evaluation would be valuable. Our focus in this paper, however, is narrower: we study how $B,S,\beta$ should scale during **pre-training under a fixed token budget**. For this question, pre-training loss is the most direct evaluation metric, as it is the quantity tied to both the theory and the optimization objective. We therefore view downstream transfer as an important follow-up, but not as necessary for validating the core claim of the paper. We will clarify this scope more explicitly in the revision. Moreover, downstream evaluations at the 1B NanoGPT scale are not standard, and such tasks are typically carried out with significantly larger models than we can pre-train under our current compute resources. We leave this to future work
>
> ### Questions
>
> Q1: This is a good question. Our formal results in the paper are stated for SCG methods, so we do not claim a fully general theorem for Adam or SGD in the current submission. However, the underlying **BST mechanism is not specific to SCG**: under a fixed token budget, one must balance a more deterministic regime (which favors larger $BS$) against a stochastic regime (where overly large $BS$ becomes suboptimal). This same mechanism already appears for SGD. Under the assumptions used in our paper—$L$-smoothness, the $\mu$-KL condition, and variance scaling $\sigma^2=\sigma_0^2/(BS)$—a standard analysis yields the recursion (up to constants) for $\Delta_k := \mathbb{E}[f(x_k)-f^\ast]$ and $\gamma \le 1/L$
> $$\Delta_{k+1}\le\Delta_k-\gamma \mu^2 \Delta_k^2+L\gamma^2 \sigma^2,$$which has the same structure as in our SCG analysis: an **optimization term** and a **noise term**. After $K$ steps, this leads to $$\varepsilon\lesssim\max\left\\{\frac{1}{K\gamma\mu^2}, \frac{\sqrt{L\gamma\sigma^2}}{\mu}\right\\}.$$Substituting $K = T/(BS)$ and $\sigma^2=\sigma_0^2/(BS)$, we obtain$$\varepsilon\lesssim\max\left\\{\frac{BS}{T\gamma\mu^2},\frac{\sqrt{L\gamma}\sigma_0}{\mu\sqrt{BS}}\right\\}.$$Optimizing in $\gamma$ gives $\gamma\asymp\min\lbrace1/L,(BS)/(T^{2/3}(L\mu^2\sigma_0^2)^{1/3})\rbrace$, and hence$$\varepsilon\lesssim\max\left\\{\frac{L\,BS}{\mu^2T},\left(\frac{L\sigma_0^2}{\mu^4 T}\right)^{1/3}\right\\}.$$Balancing these terms yields$$BS \asymp T^{2/3},$$up to problem-dependent constants. Importantly, this matches our SCG result **exactly** in form:$$BS=\left(\frac{T\mu\rho\sigma\_0}{L}\right)^{2/3},$$where in the SGD case $\rho=1$ and $(\mu,L)$ are measured in the Euclidean norm. This exact agreement is notable, as it suggests that the BST rule is not tied to a specific algorithmic structure, but reflects a more general property of stochastic optimization under a fixed token budget
>
> For Adam-type methods, the situation is more delicate. Even in classical settings, Adam is known to be non-convergent without modifications [4]. Providing a rigorous BST-type derivation for Adam would require restricting attention to convergent variants such as AMSGrad. From an algorithmic perspective, Adam-type methods interpolate between SGD/AdaGrad-like behavior and sign-based updates (which fall within the SCG family). Since both endpoints lead to the same BST scaling law, we expect the same qualitative scaling to persist in practice. We therefore view extending the formal theory to Adam as an important but non-trivial direction for future work.
>
> [4] Reddi et al, On the convergence of Adam and beyond, ICLR, 2018

---

> > ### Author Rebuttal · Reviewer_Tbqk · 2026-04-03
> >
> > I thank the authors for the clarifications. All my concerns are addressed and I maintain my score of recommending the acceptance of the paper.

---

### Decision · Program_Chairs · 2026-04-30

**Decision:**

Accept (regular)

**Comment:**

The paper studies the role of batch size in stochastic conditional gradient methods under μ-KL conditions. The authors show that there exists a critical threshold for the maximal batch size and propose an adaptive strategy that increases both batch size and sequence length. I find the results reasonable and solid, and I recommend acceptance.